# A lattice-automaton bioturbation simulator with coupled physics, chemistry, and biology in marine sediments (eLABS v0.2)

Yoshiki Kanzaki[1], Bernard P. Boudreau[2], Sandra Kirtland Turner[1], Andy Ridgwell[1,3]

[1]Department of Earth and Planetary Sciences, University of California – Riverside, Riverside, CA 92521, USA
[2]Department of Oceanography, Dalhousie University, Halifax, Nova Scotia B3H4R2, Canada
[3]School of Geographical Sciences, University of Bristol, Bristol BS8 1SS, UK

*Correspondence to*: Yoshiki Kanzaki (kanzakiy@ucr.edu)

**Abstract.** Seawater-sediment interaction is a crucial factor in carbon and nutrient cycling on a wide range of spatial and temporal scales. This interaction is mediated not just through geochemistry, but also via biology. Infauna vigorously mix sediment particles, enhance porewater-seawater exchange, and consequently, facilitate chemical reactions. In turn, the ecology and activity of benthic fauna are impacted by their environment, amplifying the sensitivity of seawater-sediment interaction to environmental change. However, numerical representation of the bioturbation of sediment has often been treated simply as an enhanced diffusion of solutes and solids. Whilst reasonably successful in representing the mixing of bulk and predominantly oxic marine sediments, the diffusional approach to bioturbation is limited by a lack of environmental sensitivity. To better capture the mechanics and effects of sediment bioturbation, we extend a published bioturbation model (acronym: LABS) by adopting a novel method to simulate realistic infaunal behavior that drives sediment mixing. In this new model (eLABS), simulated benthic organism action is combined with a deterministic calculation of water flow and oxygen and organic matter concentration fields to better reflect the physicochemical evolution of sediment in response to bioturbation. The predicted burrow geometry and mixing intensity thus attain a dependence on physicochemical sedimentary conditions. This interplay between biology, chemistry and physics is important to mechanistically explain empirical observations of bioturbation and to account for the impact of environmental changes. As an illustrative example, we show how higher organic rain can drive more intense sediment mixing by 'luring' benthic organisms deeper into sediments, while lower ambient dissolved oxygen restricts the oxic habitat depth and hence tends to reduce bulk mixing rates. Our model, with its oxygen and food availability controls, is a new tool to interpret the trace fossil record, e.g., burrows, as well as to explore biological engineering of past marine environments.

## 1 Introduction

Pore-water-particle reactions occur extensively in the upper few to hundreds of cm of marine sediments (early diagenesis), and resulting exchange with the overlying bottom water helps to regulate the chemistry of the oceans and, on relatively

longer time-scales, the concentrations of atmospheric $CO_2$ and oxygen and, thus, climate, e.g., Hülse et al. (2017). The rates of these reactions, and diagenesis overall, is influenced by benthic marine infauna, whose activities mix solid sediments and porewater solutes and modulate the exchange with overlying seawater. The consequent physical and chemical disturbance caused by infauna is defined as bioturbation *sensu lato*, e.g., Aller (1982). As sedimentary records of changing

environmental conditions are often assumed to represent monotonic changes with time, and ideally ones that can be sampled at high resolution to help understand rapid events and transitions, understanding of biota-induced physicochemical disturbances is indispensable for correct interpretation of past environmental transitions, e.g., Berger et al. (1977), Trauth (1998, 2013), Meysman et al. (2006a), Ridgwell (2007), Panchuk et al. (2008), Canfield and Farquhar (2009), Hull et al. (2011), Steiner et al. (2016), and Kirtland Turner et al. (2017). Numerical models of the appropriate processes involved

represent invaluable tools in this effort.

The transport of particles and porewater by infauna can be local and/or non-local, e.g., deposit feedings, depending on the biological properties of organisms and sediment environments, e.g., Aller (1982), and has been described as deterministic and/or stochastic processes, in one-, two- or three-dimensions, e.g., Aller (1980), Boudreau and Imboden (1987), Trauth

(1998), Shull (2001), Meysman et al. (2003, 2006b), and Reed et al. (2006). Classic models of early diagenesis, e.g., Berner (1980), Boudreau (1996, 1997), Van Cappellen and Wang (1996), and van de Velde and Meysman (2016), on the other hand, adopt relatively simple parameterizations for particle and water mixing, usually treated as diffusion (biodiffusion), because these studies focus more on the variety and effects of chemical reactions occurring within sediments than on precise/realistic description of bioturbation.

The parameterizations in these models are based on modern (current) observations, e.g., Boudreau (1994, 1998), Tromp et al. (1995), and Middelburg et al. (1997); consequently, these parameterizations may not be applicable when applied to past or future environments, which may be dominated by different groups of organisms, e.g., Savrda and Bottjer (1989), Aller (2001), Tarhan et al. (2015), Olson (2018), and van de Velde et al. (2018). Nor may they be valid if, for any given ecology,

changing environmental conditions lead to changes in the behavior, activity, or numbers of the individuals present. Note also that the benthic species responsible for bioturbation are diverse, so application of a single parameterization requires caution even for modern environmental settings (e.g., Kristensen et al., 2012).

In addition to particle mixing, benthic organisms can also modify water flow within the uppermost parts of sediments, a

process called bioirrigation (Aller, 1982). Recent modeling studies of bioirrigation include both chemical reactions and biology-induced water-exchange processes, e.g., Meysman et al. (2007) and Volkenborn et al. (2012), but not necessarily mixing of sediment particles. These latter studies also assume static burrow geometry and cannot simulate burrow development and the associated movement and metabolism of benthic organisms.

Here, we build on an existing model of animal behavior and particle mixing to create a model that simulates the coupled evolution of both burrow geometry and the physicochemical environment of sediments (Choi et al., 2002). Our model – eLABS (v0.2) – is designed to investigate the effects of biological factors on the physicochemical environment of sediments during bioturbation, or vice versa. We describe a series of experiments to illustrate model application and demonstrate how 5 our model can promote better understanding of trace fossils in the geological record.

## 2 Model overview

Our bioturbation model is a direct and traceable extension of the innovative Lattice-Automaton Bioturbation Simulator 10 (LABS) developed by Choi et al. (2002). The automaton method can suitably represent complex animal behavior with relatively simple rules, e.g., Choi et al. (2002), but also Wolf-Gladrow (2004). We refer to our version as the 'extended' LABS (eLABS) to distinguish from the original version by Choi et al. (2002), which we hereafter refer to as just LABS. We have modified the LABS FORTRAN90 code by adding deterministic calculations of oxygen and organic matter concentrations and water flow fields to improve the representation of sediment chemistry and physics. Accordingly, eLABS 15 runs two consecutive calculations: (i) a LABS simulation to account for stochastic animal behavior and sediment displacement, and (ii) the solution of a set of deterministic equations for water flow and oxygen and organic matter concentration fields in a coupled 2D diagenetic model. For a given time step, the behavior of benthic animals is simulated first, as well as associated non-local mixing of water, sediment, oxygen and organic matter. Within the same time step, information necessary for the deterministic calculations of oxygen and organic matter concentration and water flow fields is 20 collected. Then, these deterministic calculations are conducted via the coupled diagenetic model. This two-step sequence is repeated, using the oxygen and organic matter concentration fields from the previous time step as boundary conditions for the next time step simulation of organism behavior and sediment mixing. The simulation with LABS, water flow field calculation, and calculations of organic matter and oxygen concentrations are described individually in the following subsections (and see Code Availability).

### 2.1 Animal behavior

The various types of behavior of benthic organisms, and their associated impacts on sediment mixing, can be simulated in LABS, and the full details of this form of simulation were presented by Boudreau et al. (2001), Choi et al. (2002), Reed et al. 30 (2006) and Huang et al. (2007), so they are not repeated here. The essential elements of LABS are as follows:

1. A simulation occurs on a grid, i.e., a lattice, that consists of sediment, water and/or organism particles (Fig. 1). The model is continuous across the left and right edges of the lattice (Fig. 1). Particles on the grid either move or remain still according to the rules for individual particles.

2. Connected and coordinated organism particles represent infauna (Fig. 1). Individual organisms have their own pre-defined morphological properties, such as location and sizes of their heads and bodies (Fig. 1), as well as activity and gut fullness. Organisms move, push, ingest and/or egest sediment particles they encounter, depending on rules for each organism type. Behavioral rules for organisms contain probabilities for animal actions, which are resolved via randomly generated numbers and the state of both the organism and the properties of surrounding sediment and water particles (e.g., lability of sediment particle).

3. Sediment and water particles are essentially left static unless organisms move, push, ingest or egest sediment particles or unless it is time for sediment deposition or burial. The particle distribution is calculated with respect to a reference frame anchored at the (mean) sediment surface, which propagates upwards through the water column with time when there is sedimentation, i.e., the Berner diagenetic reference frame – see Berner (1980), and consequently loses particles at the base of the model during sedimentation.

4. Solid sediment particle properties include radioactive tracer content (e.g., $^{210}$Pb) or organic matter lability, while water particles have none.

A number of input parameters are required to simulate the behavior of infauna, including physical parameters to specify the sediment conditions, e.g., sedimentation rate, sediment thickness and porosity, and biological parameters to specify the characteristics of infauna, e.g., locomotion speed and size of individual animals. In our default setting (see Section 3), we employ a $12 \times 12$ cm$^2$ 2D sediment plus water grid, in which sediment bulk porosity is 0.8. The sediment-water interface is located at 3.6 cm below the top of the grid (Fig. 1), and sedimentation rate is $1.5 \times 10^{-2}$ cm yr$^{-1}$. The grid cell size is $0.05 \times 0.05$ cm$^2$ and the grid has 0.25 cm of width, such that the 2D system can be converted to a $0.25 \times 12 \times 12$ cm$^3$ 3D system (cf., Boudreau et al., 2001; Fig. 1). A single benthic animal is present with a $0.25 \times 0.25 \times 1.65$ cm$^3$ body size ($5 \times 33$ grid cells; Fig. 1), 10 cm day$^{-1}$ locomotion speed (200 grid cells day$^{-1}$), and 1 g sediment (g organism)$^{-1}$ day$^{-1}$ maximum ingestion rate (72.9 particles day$^{-1}$). The above animal properties represent those of a deposit feeder (e.g., Lopez and Levinton, 1987). Note that for the above unit conversion of ingestion rate from real to the 2D grid system, 2.5 and 1.2 g cm$^{-3}$ are assumed for the densities of sediment and organism particles, respectively. The time step for simulations with this animal is $5 \times 10^{-3}$ days (or 7.2 minutes).

As a useful feature of LABS, the simulated temporal and spatial patterns of burrows can change depending on the specific rules of individual organisms. For example, in the default setting for LABS, organisms prefer to move towards more labile organic matter. When we further impose a rule to effectively cap the organisms' consumption, such that organisms with

greater gut fullness (i.e., when they are not hungry) prefer the direction in which more water particles exist (representing a path of least resistance to travel), the resulting burrow density is lower (Fig. 2).

In LABS, the organic matter associated with a sediment particle can be parameterized with discrete lability levels. These lability levels are utilized to allow discrimination between sediment particles by organisms (e.g., Lopez and Levinton, 1987). To simulate the organic matter concentration field of sediment (see Section 2.3), we modify LABS so that each sediment particle takes a value for its organic matter concentration from a continuous distribution (from 0 to 1 wt%), instead of discrete integer levels. For convenience, changes in organic matter concentration are assumed not to change the density of a sediment particle, given the limited amount of organic matter in marine sediment (e.g., $\leq$ 1 wt% for the present study; Section 2.3). Organisms select sediment particles on the basis of organic matter concentration, assuming that the particle is more labile when it has a larger organic matter concentration (cf., Middelburg, 1989; Canfield, 1994).

Note that in LABS, a 'particle' is better thought of as a solid $0.05 \times 0.05 \times 0.25$ cm$^3$ aggregate of grains, of which a proportion of these grains can be organic matter. The size of the 'grains' comprising a solid $0.05 \times 0.05 \times 0.25$ cm$^3$ particle in the model grid is not defined but assumes to pack with no porosity. Sediment porosity is hence determined by the proportion of $0.05 \times 0.05 \times 0.25$ cm$^3$ sediment particles vs. $0.05 \times 0.05 \times 0.25$ cm$^3$ volumes of water.

Finally, we also impose rules in eLABS based on oxygen concentration in the water particles (see Section 2.3 for the calculation of oxygen concentration). For example, movement of benthic organisms can be restricted to within depths where the oxygen concentration exceeds some threshold (e.g., Huettel and Webster, 2001; Vaquer-Sunyer and Duarte, 2008). As the default setting, we impose a rule that organisms prefer to move in the direction in which the oxygen concentration is highest, i.e., organisms have low tolerance of oxygen-depleted conditions and so they avoid these conditions (Nilsson and Rosenberg, 1994). Note, however, that the stochastic behavior may occasionally lead organisms to unfavorable locations with respect to food and/or oxygen availability.

**2.2 Water flow**

Significant advective water flows can be caused by infauna within sediments when they move, and/or push, ingest and/or egest sediment particles. Non-local mixing of water accompanying displacements of sediment particles by infauna is already represented in LABS (see above). In eLABS, we further implement a deterministic calculation of water flow-field, which accounts for the advective flows caused by organisms. We assume that sediment particles are impermeable (see above), and the presence of animals does not block the water flows they cause (Meysman et al., 2007; Volkenborn et al., 2012), i.e., we treat organism particles in the same way as water particles. Then, the system is binary with respect to fluid flow. To

calculate a water flow-field, we solve the Navier-Stokes equation by the marker and cell method, defining the water flow velocities $\mathbf{u}$ (cm yr$^{-1}$) and pressures $p$ (g cm$^{-1}$ yr$^{-2}$) on the edges and centers of the eLABS grid cells, respectively (Harlow and Welch, 1965; Hoffmann and Chiang, 2000; Manwart et al., 2002):

$$\frac{\partial \mathbf{u}}{\partial t} + (\mathbf{u} \cdot \nabla)\mathbf{u} = -\frac{1}{\rho}\nabla p + \nu \nabla^2 \mathbf{u} \qquad (1)$$

Here, $t$ is time (yr), and $\rho$ and $\nu$ are respectively the density and kinematic viscosity of water (1.00 g cm$^{-3}$ and $4.79 \times 10^5$ cm$^2$ yr$^{-1}$ at 5 °C; Kestin et al., 1978). The symbols of $\nabla$ and $\nabla^2$ represent the vector differential and Laplace operators, respectively. We assume negligible external forces in Eq. (1). At the top and bottom layers, we impose no-vertical-flux

(i.e., zero pressure gradients) boundary conditions (cf., Meysman et al., 2005; Volkenborn et al., 2012) and left and right boundaries are periodic (Section 2.1). No-slip boundaries (i.e., zero velocities) are assumed at interfaces between sediment and water/organism particles. When organisms displace sediment particles, constant flows are imposed at the middle of the head or tail of the organisms, reflecting the velocities of moving organism and sediment particles so that momentum is conserved. Note that, given the assumptions and boundary conditions above, the flow calculation in the present study may

not be appropriate for permeable sediments, e.g., Huettel and Webster (2001). Nonetheless, the relatively fast rates of flows above the seawater-sediment interface are accounted for by considering eddy diffusion (see Section 2.3). Approximate steady-state $\mathbf{u}$ is obtained by solving Eq. (1) with time steps of 0.025 seconds until change becomes insignificant with time, which usually requires less than one model second. Figure 3 illustrates an example stream function caused by a benthic organism at 225 model days after the start of an eLABS simulation with default settings (Section 3).

## 2.3 Oxygen and organic matter

In LABS, a water particle has no specific physicochemical properties. In eLABS, however, water particles have individual oxygen concentrations. Sediment particles are assumed to have negligible oxygen (cf., Volkenborn et al., 2012). Organism

particles are treated in the same way as water particles, i.e., they have individual oxygen concentrations. The system is then binary (sediment vs. water/organism particles) with respect to oxygen concentration. The calculation of the oxygen concentration is conducted on a grid that is occupied by water/organism particles with a general advection-diffusion-reaction equation (e.g., Boudreau, 1997):

$$\frac{\partial[O_2]}{\partial t} = \nabla \cdot (D\nabla[O_2] - \mathbf{u}[O_2]) - R \qquad (2)$$

Here, $[O_2]$ is the dissolved oxygen concentration (mol $L^{-1}$), $D$ is the effective diffusion coefficient ($cm^2$ $yr^{-1}$), which accounts for eddy diffusion above seawater-sediment interface, as well as molecular diffusion, and $R$ represents the oxygen consumption rate by aerobic decomposition of organic matter and biological respiration (mol $L^{-1}$ $yr^{-1}$). Note that biological oxygen sources, which may be important for sediments in nearshore areas (e.g., Jahnke, 2001), are not considered in Eq. (2).

We adopt the following formulation for the effective diffusion coefficient $D$ (Boudreau, 2001; Volkenborn et al., 2012):

$$D = D_0 + 0.4\nu(zu^*/\nu)^3/363 \qquad (3)$$

10    where $D_0$ is the molecular diffusion coefficient ($3.88\times10^2$ $cm^2$ $yr^{-1}$ at 5 °C; Schulz, 2006), $z$ is the height above seawater-sediment interface (cm), $u^*$ is the shear velocity and $\nu$ is again the kinematic viscosity of water. In our default setting, we assume that $u^* = 1.0\times10^6$ cm $yr^{-1}$ (cf., Pope et al., 2006; Volkenborn et al., 2012). The oxygen consumption term $R$ is given by

15    $$R = (k_{rsp} + k_{dcy})m[O_2] \qquad (4)$$

where $m$ is the concentration of organic matter in sediment particles (wt%), while $k_{rsp}$ and $k_{dcy}$ (wt%$^{-1}$ $yr^{-1}$) are the apparent rate constants for the biological respiration and aerobic decomposition of organic matter, respectively. Note that it has been reported that the rate of organic matter oxidation can be independent of oxygen concentration (e.g., Jørgensen and Boudreau, 20    2001), though the mechanisms that explicitly explain the oxygen dependence of organic matter oxidation in sediments are not yet fully understood (cf., Hulthe et al., 1998; Dauwe et al., 2001; Archer et al., 2002; Arndt et al., 2013).

In the present study, we assume first-order dependence for Eq. (4). In our default setting, $k_{dcy} = 4.54\times10^2$ wt%$^{-1}$ $yr^{-1}$ and $k_{rsp} = 10^4\times k_{dcy}$. Note that this $k_{dcy}$ value corresponds to a pseudo first-order decay constant for organic matter of 0.1 $yr^{-1}$ (cf., 25    Canfield, 1994), if $[O_2]$ is constant at $2.2\times10^{-4}$ mol $L^{-1}$. At individual interfaces between sediment and water/organism particles and the bottom layer of the grid, impermeable (i.e., zero concentration gradients) boundary conditions are imposed. The constant oxygen concentration ($2.2\times10^{-4}$ mol $L^{-1}$ in the default setting; Volkenborn et al., 2012) is assumed at the top layer of the grid as another boundary condition. The default initial condition for the calculation of oxygen concentration is $[O_2] = 2.2\times10^{-4}$ mol $L^{-1}$ for every water/organism particle.

Our calculations are conducted using an implicit finite-difference method, adopting the first-order upwind and second-order central differencing schemes for the first- and second-order spatial derivatives, respectively, in Eq. (2). The calculation proceeds by first collecting flow velocity data (**u**) from the water flow calculation (Section 2.2), then determining the

effective diffusion coefficients and consumption rates in the individual finite difference grid cells (i.e., grid cells occupied with water/organism particles) based on Eqs. (3) and (4), respectively, and finally solving the finite difference version of Eq. (2), reflecting the **u**, $D$ and $R$ values. See below for more details on the calculation of the oxygen consumption term $R$. Mass conservation of oxygen has been verified even with temporal changes of boundary conditions as a result of particle

5    displacements by benthic animals (Appendix A).

    The concentration of organic matter in sediment particles that are not consumed by benthic organisms decreases with time, according to the following equation:

$$\frac{\partial m}{\partial t} = -\gamma k_{dcy} m[O_2] \tag{5}$$

Here, $\gamma$ is the unit conversion factor from mol L$^{-1}$ to wt% (= 1.2 wt% mol$^{-1}$ L, assuming that sediment particle has 2.5 g cm$^{-3}$ for density, negligible porosity and CH$_2$O for organic matter chemical formula). For a sediment particle that is consumed by benthic animals, the concentration decreases faster:

$$\frac{\partial m}{\partial t} = -\gamma (k_{rsp} + k_{dcy}) m[O_2] \tag{6}$$

Note that, although not described by Eqs. (5) and (6), organic matter is mixed by organisms and deposited and buried along with sediment particles (as in LABS — see the first paragraph of Section 2 and Section 2.1).

    As an initial condition, organic matter concentrations are randomly assigned to individual sediment particles in the range of $\leq 1$ wt%, with the probability of high concentration decreasing with depth in the default setting. The sediment particles deposited to the seawater-sediment interface are assumed to have 1 wt% of organic matter (e.g., Müller and Suess, 1979), as a default setting. The above default initial and boundary conditions are adopted to save time to reach steady state (e.g., Van

25    Cappellen and Wang, 1996). The distributions of organic matter evolve from the above initial condition, following Eqs. (5) and (6), while occasionally being added and lost with sediment particles at times of deposition and burial, respectively. The calculation of organic matter concentration is conducted by an explicit finite-difference method. The degradation rate of each sediment particle is first obtained as sum of degradation rates, calculated from Eqs. (5) and (6), with the particle's organic matter concentration and oxygen concentrations of the surrounding neighbor cells occupied with water/organism

30    particles (4 neighbor cells at maximum). At the same time, data on degradation rates corrected by a factor of $-\gamma$ are passed to these neighbor water/organism particles as $R$ values, Eqs. (3)–(6), to be used in the oxygen calculation (see above). The organic matter concentration in a given time step is then obtained by subtracting the calculated degradation rate, multiplied

with the time step ($5 \times 10^{-3}$ days, Section 2.1), from the organic matter concentration in the previous time step (Eqs. (5) and (6)).

## 3 Results and discussion

Example results with eLABS are discussed in this section. First, we consider the effects of several biological, physical and chemical factors on bioturbation on a relatively short time scale (1 model year, Section 3.1). Then, we examine temporal changes in biological and sedimentary physicochemical interactions during bioturbation by extending the simulation duration to 10 model years (Section 3.2). Finally, we illustrate the utility of eLABS for theoretical prediction of bioturbation
10    effects with simulations for an ocean with a low oxygen concentration (Section 3.3). The model parameterizations for these simulations are summarized in Table 1. The effects of the calculation domain size and resolution for the water flow field on the results shown in this section are likely insignificant (Appendix B).

### 3.1 One-year simulations

eLABS can yield the time evolution of oxygen and organic matter concentration profiles (e.g., Fig. 4) and fluxes of oxygen from the sediment-water interface (e.g., Fig. 5), in addition to the time evolution of burrow geometry (e.g., Fig. 6) and the biodiffusion coefficient ($D_b$) (e.g., Fig. 7), obtained through the LABS part of the calculation. We can also calculate the relative change in sediment permeability and formation factor, which is related to tortuosity (Boudreau, 1997; Clennell,
1997), as functions of time through the reactive transport part of our calculation (Appendix C); such results are useful to the investigation of feedbacks between biological, chemical and physical factors during bioturbation.

To illustrate these feedbacks between chemistry, physics, and biology, we compare results from six simulations (Table 1): (a) a simulation with the default settings, (b) a case that assumes that organisms have tolerance to oxygen-depleted
conditions but otherwise default settings, (c) a case where the aerobic decomposition rate-constant for organic matter is ten times higher, with otherwise default settings (i.e., $k_{dcy} = 4.54 \times 10^3$ wt%$^{-1}$ yr$^{-1}$ and $k_{rsp} = 10^3 \times k_{dcy}$), (d) a case where sediment bulk porosity is lowered to 0.6, with otherwise the default settings, (e) a case that assumes ten times larger shear velocity ($u^*$ = $1.0 \times 10^7$ cm yr$^{-1}$), with otherwise the default settings, and (f) a case without deterministic water flow, with otherwise default settings (Figs. 4–7). All six simulations were conducted for one model year (Table 1). The effects of biological,
chemical and physical parameters on bioturbation on this relatively short time scale are illustrated by comparing simulations (b) through (f) with simulation (a). Each simulation was run five times to account for the effect of stochastic animal

behavior during bioturbation. We report only one representative result from the five results for each simulation and denote the stochastic effect, e.g., result diversity, if significant. See Appendix D for the details on the contribution of stochastic animal behavior to bioturbation.

Before making comparisons between simulations, we discuss overall flux and biodiffusion coefficient results (Figs. 5 and 7). The total oxygen consumption flux is calculated as the sum of oxygen consumption flux by aerobic degradation of organic matter and the infaunal respiration flux, i.e., Eq. (4). The calculated total consumption fluxes (black dotted curves, Fig. 5) are within the ranges observed in various (from deep to coastal) marine sediment settings in the present oceans (e.g., ~10 to $10^3$ $\mu$mol cm$^{-2}$ yr$^{-1}$, Jahnke, 2001; Meile and Van Cappellen, 2003), suggesting that our model is a reasonable

simulation of reality. Oxygen consumption by respiration has been reported to be a few to several tens of $\mu$mol cm$^{-2}$ yr$^{-1}$ in laboratory experiments, which contain $1\times10^2$ to $2\times10^3$ individuals of infauna, with body size comparable to or smaller than that assumed in the present study per m$^2$ (Cammen, 1980; Kemp, 1987). These observations are consistent with calculated values for the respiration flux in the present study, i.e., mostly less than a hundred $\mu$mol cm$^{-2}$ yr$^{-1}$ as an annual average for simulations with $3.3\times10^3$ m$^{-2}$ population (with one exception of 180 $\mu$mol cm$^{-2}$ yr$^{-1}$). Note that infaunal respiration flux is

not explicitly shown in the present study (e.g., Fig. 5) but can be calculated simply as the difference between the total O$_2$ consumption flux and the O$_2$ flux by aerobic decomposition of organic matter, i.e., black dotted curves minus turquoise curves in Fig. 5. Deviations of diffusive oxygen fluxes from total oxygen consumption fluxes represent time changes of total O$_2$ amount in the calculation domain, which are significant when porewater chemistry is far away from steady state (<dozens of days from the start of simulation, Fig. 5). See Appendix A for more details on the oxygen fluxes.

Biodiffusion coefficients ($D_b$) obtained in the present study (e.g., Fig. 7) are also consistent with observed values, e.g., $10^{-3}$ to $10^2$ cm$^2$ yr$^{-1}$ at around $10^{-2}$ cm yr$^{-1}$ burial velocity (Boudreau, 1994). Note that we show values of the biodiffusion coefficient only for simulation (a), denoted as $D_{b,std}$ in Fig. 7a. For other non-default simulations (b–f), we show ratios of biodiffusion coefficients to those obtained in the default run ($D_b/D_{b,std}$) to facilitate comparison. Also note that the

biodiffusion coefficients in the present study are obtained by calculating average values of squared displacements of individual sediment particles divided by four times the time required to make the displacements (cf., Boudreau et al., 2001). Furthermore, the displacements by sediment burial are not counted in the above displacement calculation and thus not reflected in the biodiffusion coefficients. The calculation of displacements is conducted at different depths and thus the biodiffusion coefficient is reported as a function of sediment depth as shown in Fig. 7 (Reed et al., 2006).

Comparison of simulations (a) and (b) (panels (a) and (b) of Figs. 4–7) illustrates one example of oxygen's effect on bioturbation. Benthic organisms may possess an oxygen sensor because of the fatal effects of oxygen depletion, while other organisms may adapt to oxygen removal and hence not require one (cf., Vaquer-Sunyer and Duarte, 2008). For example,

Nilsson and Rosenberg (1994) experimentally examined responses of macrobenthos (including echinoids, bivalves, ophiuroids, polychaetes and holothuroidea) to hypoxia and reported that most of the examined species responded by escaping oxygen-depleted sediments; however, some species (the polychaetes *Nephtys incisa* and *N. hombergii*) showed greater tolerance and stayed longer in hypoxic sediments than others.

In the default setting, when the organism has little tolerance to oxygen depletion, it cannot go deeper than ~4 cm into sediments (Fig. 6a) because of the limited oxygen penetration (Fig. 4a). In contrast, when the organism has tolerance to hypoxia, i.e., simulation (b), it can move into deeper sediments despite the relative oxygen depletion (Figs. 4b and 6b). The result is deeper burrows in simulation (b) compared to (a), which facilitates oxygen transport into sediments, e.g., compare

250-days results between Figs. 4a and 4b. Nevertheless, oxygen consumption fluxes by aerobic decomposition of organic matter are similar between simulations (b) and (a), i.e., 30 vs. 28 $\mu$mol $O_2$ cm$^{-2}$ yr$^{-1}$ at 1 year. The effects of biological response to $O_2$ depletion on these oxygen consumption fluxes are mitigated because we assume that organic matter concentration decreases with depth initially and burrow geometry differs significantly between (a) and (b) only deep within the sediments. Similarly, annual average infaunal respiration does not significantly differ between simulations (a) and (b),

21 vs. 19 $\mu$mol $O_2$ cm$^{-2}$ yr$^{-1}$ (difference between total consumption and aerobic OM decomposition from each of Figs. 5a and 5b). Note that biological respiration occurs in pulses, and it is meaningful only if we compare time-averaged fluxes for respiration. We thus evaluate the organic matter decomposition flux values at the end of simulations and the infaunal respiration flux averages over the entire simulation. In simulation (a), the calculated $D_b$ value is smaller than that in simulation (b) at depths deeper than ~6 cm (Fig. 7b). These differences are attributed to the low organism tolerance to

hypoxia in simulation (a), which restricts sediment mixing to shallow oxygen-rich sediments, as reflected in burrow geometry in Figs. 4 and 6.

Comparison of simulations (c) and (a) provides insight into the effect of greater organic matter reactivity. Because of the increase in oxygen consumption by aerobic decomposition of organic matter, oxygen cannot penetrate deeper than ~3 cm in

simulation (c) (Fig. 4c), which also restricts the movement of the organism because of its low tolerance to oxygen depletion (Fig. 6c). Accordingly, oxygen consumption fluxes are comparable in simulations (a) and (c) despite the order of magnitude difference in the apparent rate constant, i.e., 28 vs. 54 $\mu$mol $O_2$ cm$^{-2}$ yr$^{-1}$ at 1 year for aerobic organic matter decomposition and 21 vs. 37 $\mu$mol $O_2$ cm$^{-2}$ yr$^{-1}$ as annual averages for infaunal respiration (Figs. 5a and 5c). The limited effect on oxygen consumption fluxes is also partially attributable to the assumed decrease of organic matter concentration with depth, which

mitigates the effects of different sediment geometries, especially at greater depths, and also to the decrease in organic matter concentration through decomposition at shallower depths. Because sediment mixing is limited to shallow parts of sediments, the biodiffusion coefficient in simulation (c) is significantly lower than that in (a) at depths deeper than ~4 cm (Fig. 7c).

However, the extent of reduction in sediment mixing at depth in (c) relative to (a) can vary because burrow development can be affected by the stochastic animal behavior as well (cf., Appendix D).

We can examine the potential effects of decreased porosity on bioturbation by comparing the simulation results between (a) and (d). When porosity is decreased, oxygen penetration shallows (Figs. 4a and 4d) because of increased tortuosity and lowered permeability and because the number of particles, and thus organic matter, per unit volume is higher. Accordingly, high burrow density is observed only in the shallow regions in simulation (d), as the organism avoids oxygen-depleted conditions (Fig. 6d). Despite the shallower penetration of oxygen and development of fewer deep burrows, oxygen fluxes in simulation (d) are not significantly lower than simulation (a), i.e., 26 vs. 28 $\mu$mol $O_2$ cm$^{-2}$ yr$^{-1}$ at 1 year for aerobic decomposition of organic matter and 22 vs. 21 $\mu$mol $O_2$ cm$^{-2}$ yr$^{-1}$ as annual averages for infaunal respiration (Figs. 5a and 5d). Again, this can be attributed to the increased amount of organic matter per unit volume with decreased porosity. As the bioturbated zone is relatively shallow in simulation (d), the biodiffusion coefficient is significantly reduced at ~5 to 8 cm depths, compared to (a) (Fig. 7d). Again, the details of the relative difference in biodiffusion coefficient between (d) and (a) can vary among different runs as a result of the stochastic animal behavior (cf., Appendix D). We further note that the extent of oxygen penetration and the biodiffusion profiles are similar between simulations (d) and (c) (Figs. 4c, 4d, 7c and 7d), whereas the total oxygen consumption flux and burrow geometry are significantly different between the two simulations (Figs. 5c, 5d, 6c and 6d).

Next, oxygen can penetrate deeper into sediments when shear velocity is higher (simulation (e)) (Figs. 4a and 4e). A higher shear velocity causes enhanced oxygen penetration into the sediment because of a stronger turbulent flux to the sediment-water interface with increased eddy diffusion, i.e., Eq. (3) of Section 2.3. With increased oxygen penetration, the annual average oxygen consumption by infaunal respiration is larger in simulation (e) than (a), i.e., 180 vs. 21 $\mu$mol $O_2$ cm$^{-2}$ yr$^{-1}$ (Figs. 5a and 5e). In turn, oxygen consumption by aerobic organic matter decomposition is smaller in simulation (e) than (a), 23 vs. 28 $\mu$mol $O_2$ cm$^{-2}$ yr$^{-1}$ at 1 year (Figs. 5a and 5e). The lower aerobic organic matter decay in simulation (e) is attributed to significantly higher respiration, which dominates consumption of organic matter. With deeper oxygen penetration, the burrow density, as well as the biodiffusion coefficient at deep depths, is higher in simulation (e) than (a) (Fig. 7e). However, the extent to which the biodiffusion coefficient and burrow development are enhanced in deep sediments can change between repeated runs due to the stochastic animal behavior (cf., Appendix D). Note that simulations (e) and (b) are relatively similar with respect to the biodiffusion coefficient (Figs. 7b and 7e), although different in terms of oxygen profiles and fluxes and burrow development (Figs. 4b, 4e, 5b, 5e, 6b and 6e). One can also obtain rather similar bioturbation results to those in simulation (e) by arbitrarily increasing biologically induced water flows (Appendix E).

When we remove the advective flow from the calculations, i.e., simulation (f), the resultant oxygen profiles, fluxes, burrow geometry and biodiffusion coefficient are generally similar to those with the default settings (panels (a) and (f) of Figs. 4–7). Accordingly, with the assumptions adopted in the present study, advective water flow accompanying animal movements and sediment displacements influences bioturbation insignificantly. See simulation (e) above and Appendix E for the cases where advective water transport is implicitly enhanced by ocean currents and benthos, respectively.

## 3.2 Ten-year simulations

To assess the impact on bioturbation over time scales approaching those characteristic of shallow-water, high-deposition, marine environments, we provide in this subsection 4 additional model simulations run for 10-years each (Figs. 8–11 and Table 1). All the simulations assume porosity of 0.6, which enables us to detect easily the effects of greater run time on the resultant burrow geometry and biodiffusion coefficient (Figs. 6 and 7). All simulations were run without the water flow calculation to reduce the computational effort.

Specifically, we conducted simulations with (a) the default settings except for lower porosity and minus water flow (see above), (b) assuming that organisms have tolerance to hypoxia, (c) 10× higher decay constant for organic matter but otherwise the same settings as in (a), and (d) 10× higher sedimentation rate (0.15 cm yr$^{-1}$) but otherwise the same settings as in (a) (panels (a)–(d), respectively, of Figs. 8–11; Table 1). As in Section 3.1, we compare simulations (b) through (d) with simulation (a) to show the effects of individual parameter variations on bioturbation. Note that the biodiffusion coefficients in simulations (b) to (d) are shown only relative to those in (a) and absolute values of biodiffusion coefficients are given only for (a) (Fig. 11). Each case was simulated 5 times. Only one representative result is presented for each case, and the stochastic effects are noted only when significant.

Comparisons of simulations (b) and (c) with (a) suggest that some of the parameter influences described in Section 3.1 hold on longer time scales but others may not. With the addition of tolerance to oxygen depletion, i.e., simulation (b), the organism can penetrate deeper sediments than in (a), resulting in correspondingly deeper burrows and increased sediment mixing (Figs. 10a, 10b and 11b), but not necessarily further oxygen penetration (Figs. 8a and 8b) depending on the structure of burrows and stochastic animal behavior. Also, oxygen consumption by infaunal respiration in simulation (b) is not necessarily larger than that in (a), 14 vs. 15 $\mu$mol $O_2$ cm$^{-2}$ yr$^{-1}$ as 10 years averages (Figs. 9a and 9b), probably as a result of the similar residence time of the organism in $O_2$ enriched sediments.

Aerobic decomposition of organic matter decreases with time more significantly in (a) than (b), and results in modestly lower $O_2$ flux, e.g., 13 vs. 15 $\mu$mol $O_2$ cm$^{-2}$ yr$^{-1}$ at 10 years (Figs. 9a and 9b). The time change in the $O_2$ flux via organic

matter decomposition in simulation (a) is likely caused by the fact that the organisms in (a) are limited to shallow depths at first and consume organic matter close to the seawater-sediment interface. As time passes (> 1 year), the surface sediment gets depleted in organic matter and porewater becomes oxygenated, so that the location where decomposition of organic matter dominantly occurs shifts to deeper depths; this results in smaller rates of organic matter decomposition overall, given the assumed initial depth-dependence of organic matter concentration (Section 2.3). In contrast, the tolerance to hypoxia allows the organism in (b) to mix sediment more homogenously (i.e., with lateral and vertical mixing of similar intensity) despite $O_2$ conditions, resulting in a more time-invariant $O_2$ flux (Fig. 9b).

With higher reactivity of organic matter, i.e., simulation (c), oxygen penetration is so shallow that the organism cannot dig into sediments but mostly bulldozes the uppermost sediment particles for greater than 50 days (Figs. 8c and 10c). Accordingly, burrow development is limited to shallow sediment and sediment mixing is weaker at depth (Figs. 10a, 10c and 11c) (cf., Section 3.1). However, with elapsed time (> 1 yr) and due to the stochastic animal behavior, burrows can generate a structure that allows efficient oxygen penetration, and the biodiffusion coefficient becomes comparable to that in (a). $O_2$ consumption fluxes are comparable between simulations (c) and (a), e.g., 16 vs. 13 $\mu$mol $O_2$ cm$^{-2}$ yr$^{-1}$ at 10 years for aerobic organic matter decomposition (Figs. 9a and 9c), and 14 vs. 15 $\mu$mol $O_2$ cm$^{-2}$ yr$^{-1}$ as 10-year averages for infaunal respiration (Figs. 9a and 9c), consistent with the 1-year simulations (Section 3.1).

The effect of high sedimentation rate, which is difficult to see in 1-year simulations (Section 3.1), can be examined by comparing simulations (d) and (a). Sediment in (d) is less oxygenated than (a) because of the higher organic matter supply to the system (Figs. 8a and 8d). Because sediment particles with 1 wt% organic matter rain 10 times more frequently (Section 2.3), $O_2$ consumption fluxes are higher and remain more constant for 10 years in (d) than (a), e.g., 25 vs. 13 $\mu$mol $O_2$ cm$^{-2}$ yr$^{-1}$ at 10 years for aerobic decomposition of organic matter and 16 vs. 15 $\mu$mol $O_2$ cm$^{-2}$ yr$^{-1}$ as 10 years averages of infaunal respiration (Figs. 9a and 9d). Interestingly, sediment mixing and burrow development extend deeper into sediment in simulation (d) than (a) (Figs. 10a, 10d and 11d). We attribute the mixing enhancement in (d) to the higher frequency at which sediment particles with 1 wt% organic matter rain into burrows, which lures the organism deeper into the sediment despite less-oxygenated conditions. Modern observations have suggested greater mixing by infauna in sediment that receives larger organic matter rain, e.g., Berger and Killingley (1982), Boudreau (1994, 1998), Tromp et al. (1995), Trauth et al. (1997), and Archer et al. (2002). One of mechanistic explanations for this relationship can thus be given by the present model, i.e., the deposition of relatively fresh sediment particles into infaunal burrows.

### 3.3 Low-$O_2$ simulations

Our model provides the ability to explore potential impacts on bioturbation of animal behavior under conditions that may occur or have occurred in the future/past, but cannot directly be observed through experiments. To illustrate such an application, we consider 4 additional simulations assuming a past ocean with less oxygen (e.g., Lu et al., 2018). The four simulations in this subsection are parameterized in the same way as those in Section 3.2, except that the oxygen

concentration at the upper boundary is taken as $0.1\times$ that in the previous simulations (i.e., $2.2\times10^{-5}$ mol $L^{-1}$). The four simulations assume (a) the default settings except for porosity, water flow and oxygen concentration (see just above and Section 3.2), (b) additionally that organisms have tolerance to hypoxia, (c) $10\times$ higher decay constant for organic matter but otherwise the same settings as in (a), and (d) $10\times$ higher sedimentation rate (0.15 cm $yr^{-1}$) but otherwise the same settings as in (a) (panels (a)–(d), respectively, of Figs. 12–15; Table 1). As in Sections 3.1 and 3.2, we compare simulations (b) through

(d) with simulation (a) and describe the effects of individual parameter variations on bioturbation. Only one representative result is presented from five runs for each simulation as in the previous subsections.

Tolerance to hypoxia allows deep burrows to develop and, consequently greater sediment mixing in simulation (b), as expected from Sections 3.1 and 3.2 (Figs. 14a, 14b and 15b). Oxygen consumptions by aerobic decomposition of organic

matter are similar between simulations (a) and (b), e.g., 3.1 vs. 3.3 μmol $O_2$ $cm^{-2}$ $yr^{-1}$ at 10 years (Figs. 13a and 13b), while infaunal respiration can be lower in simulation (b), 1.0 vs. 2.5 μmol $O_2$ $cm^{-2}$ $yr^{-1}$ as 10-year averages (Figs. 13a and 13d), consistent with the descriptions in Section 3.2. Note that distributions of normalized oxygen concentration are not significantly altered by changing the oxygen concentration at the upper boundary (Figs. 8 and 12); given our assumed organic matter decomposition kinetics, the equation for normalized oxygen concentration remains the same despite the

change in the boundary oxygen concentration (Eqs. (2) and (4)).

The effect of assuming high reactivity for organic matter, i.e., comparing panels (c) with (a) of Figs. 12–15, is generally consistent with that described in Sections 3.1 and 3.2. However, depending on the stochastic animal behavior and the burrow geometry, the effect of less oxygen penetration in (c) than (a) can vary between different runs. Oxygen consumption

via aerobic decomposition of organic matter is significantly higher in (c) than (a), 13 vs. 3.1 μmol $O_2$ $cm^{-2}$ $yr^{-1}$ at 10 years (Figs. 13a and 13c), while respiration fluxes are relatively similar between simulations (a) and (c), 2.5 vs. 2.0 μmol $O_2$ $cm^{-2}$ $yr^{-1}$ as 10-year averages (Figs. 13a and 13c). The lower respiration contribution to oxygen consumption in simulation (c) of this subsection compared to that in Section 3.2 is attributed to the longer residence of the organism in less oxygenated sediment.

High sedimentation results in shallower oxygenation of sediment (Figs. 12a and 12d) and similar oxygen consumption fluxes, e.g., 3.6 vs. 3.1 μmol $O_2$ $cm^{-2}$ $yr^{-1}$ at 10 years for aerobic decomposition of organic matter and 1.3 vs. 2.5 μmol $O_2$ $cm^{-2}$ $yr^{-1}$ as 10-year averages for infaunal respiration (Figs. 13a and 13d). With high sedimentation, burrow development

and sediment mixing are deeper (Figs. 14a, 14d and 15d), consistent with descriptions in Section 3.2. Similar results between Sections 3.2 and 3.3 suggest that the availability of food rather than oxygen dominantly determines the preferable direction for the organism's movement under less oxygenated conditions. Accordingly, with the assumptions in simulation (d), the modern empirical relationship between biological mixing intensity and rain rate of organic matter (e.g., Berger and Killingley, 1982; Boudreau, 1994; Tromp et al., 1995; Trauth et al., 1997; Archer et al., 2002; Section 3.2) might still hold in the 10× less oxygenated oceans.

## 4 Summary and Conclusions

Here, we present an extension to the original LABS model of animal behavior and sediment mixing (Choi et al., 2002) to include dissolved oxygen distributions in marine sediments and their influence on the biological processes – 'eLABS'. The results from eLABS reveal the existence of complex inter-related effects of biological, chemical and physical parameters on oxygen fluxes and rates of mixing in ocean sediments. The effects of these variations are not straightforwardly reflected in the oxygen consumption fluxes, burrow development, or sediment mixing. However, we note that our example simulations consider only a limited range of variability within the full parameter space. Boudreau et al. (2001) examined variations in other biological parameters, e.g., number of infauna, locomotion speed and ingestion rate of individual organisms, and found large effects using LABS. We believe that eLABS would predict similar effects on bioturbation with variations in these same biological parameters (not tested explicitly here), but eLABS extends the ability to consider impacts on oxygen concentration and burrow geometry.

The eLABS model is useful for theoretical investigations into the interplay between biological, physical and chemical factors influencing sediment bioturbation. Our goal is ultimately to provide a mechanistic explanation for empirical relationships observed in the modern ocean sediments between bioturbation and other sediment properties and processes. Such a mechanistic understanding will be particularly useful for interpreting the extent to which bioturbation has modified geological records of past environmental events. This study has shown the above goal and application are feasible with our new model.

Further development of eLABS will consider a population of organisms that can vary in size depending on food availability, competition for that food and predation [Kanzaki et al., in prep.]. Further useful extensions to eLABS include anaerobic degradation of organic matter, increased flexibility and applicability of the water flow field calculations (e.g., application to water-pumping actions by infauna; Meysman et al., 2005; Appendix E), and increased overall calculation efficiency to enable longer run times and/or a deeper sediment column. An experimental study, whose details can be

compared with the model's settings and results, is also desirable to further confirm our model's validity (e.g., Volkenborn et al., 2012).

## Code Availability

The source codes of the extended LABS (eLABS) model are available on GitHub (https://github.com/imuds/iLABS) under the GNU General Public License v3.0. The specific version used of the eLABS model in this paper is tagged as 'eLABSv0.2' and has been assigned a DOI: 10.5281/zenodo.3451420. The version of eLABS at the first submission of this paper and the original LABS codes are also available, tagged as 'eLABSv0.1' (assigned a DOI: 10.5281/zenodo.3451416)
and 'original_LABS' (DOI: 10.5281/zenodo.3451415), respectively. A readme file on the web provides the instructions for executing the model and plotting the results.

## Appendix A: Conservation of oxygen

The fluxes of oxygen calculated in eLABS include those from changes in the total oxygen amount in the calculation domain, aerobic degradation of organic matter, infaunal respiration, water advection and eddy plus molecular diffusion. In addition, the residual flux of oxygen as a sum of the above oxygen fluxes is computed to confirm the conservation of oxygen, i.e., the residual oxygen flux must be close to zero if oxygen is conserved within the calculation domain. Note that oxygen flux is positive when oxygen is supplied to the calculation domain, while it is negative when lost from the sediment. In Section 3,
we report only the absolute flux values of total oxygen consumption, aerobic OM decomposition, and diffusive oxygen supply. An example of a detailed flux calculation, broken down into all the above-mentioned fluxes, is shown in Fig. A1a for a simulation with the default settings. The residual flux of oxygen is insignificant in all simulations throughout the calculation time (e.g., Fig. A1b). Mass conservation is satisfied even with temporal changes in boundary conditions through displacement of sediment particles by infauna, as confirmed from a simulation without oxygen consumption and
ingestion/egestion (Fig. A1c). Note that when implementing ingestion by infauna, sediment particles are temporarily removed from the calculation domain and replaced by water particles with zero oxygen concentration. After this replacement, oxygen transport flux toward the replaced water particles is induced, which physically makes sense and is relatively insignificant (~1 % of oxygen consumption by aerobic organic matter degradation and infaunal respiration).

## Appendix B: Calculation domain size and grid resolution for water flow simulation

Two simulations with a wider sediment size ($24 \times 12$ cm$^2$) and a finer grid resolution for water flow calculation ($480 \times 480$) were conducted (Figs. B1–4), and the results suggest that the default domain size and resolution for the fluid flow simulation are sufficient for generating reasonable bioturbation results. Note that the number of benthos is increased in the former simulation to maintain the population per unit area of sediment ($3.3 \times 10^3$ m$^{-2}$).

## Appendix C: Permeability and formation factor

The water flow field calculation can be applied to evaluate relative permeability changes through burrow development. By imposing a hypothetical pressure difference $\Delta p$ (g cm$^{-1}$ yr$^{-2}$) over the length $L$ of the eLABS grid and obtaining a corresponding steady-state water flux $q$ (g cm$^{-2}$ yr$^{-1}$) through the grid (e.g., Fig. C1), the permeability $\kappa$ (cm$^2$) can be calculated as $\kappa = \nu q L / \Delta p$ where $\nu$ is the kinematic viscosity of water ($4.79 \times 10^5$ cm$^2$ yr$^{-1}$). According to the calculated relative changes in $\kappa$, permeability generally increases with burrow development as illustrated in Fig. C2.

Similarly, the oxygen calculation in eLABS can be utilized for estimating relative changes in the formation factor, by considering only the diffusion term and assuming steady state and a constant effective diffusion coefficient; then the governing equation (i.e., $D\nabla^2[O_2] = 0$) becomes the same as that for electric potential $\Phi$ (V), i.e., $\sigma\nabla^2\Phi = 0$, where $\sigma$ is the electrical conductivity ($\Omega^{-1}$ cm$^{-1}$) (e.g., Klinkenberg, 1951). By applying a hypothetical electric potential difference $\Delta\Phi$ (V) over the length $L$ of the eLABS grid and solving for $\Phi$ in individual grid cells after replacing $[O_2]$ by $\Phi$ and $D$ by $\sigma$ (e.g., Fig. C3), the electric current $J$ through the grid (A cm$^{-2}$) can be calculated as $J = -\sigma\partial\Phi/\partial y$ where $y$ is the length (cm) in the direction parallel to the direction in which $\Delta\Phi$ is imposed. Then, the formation factor $F$ can be obtained as $F = -\sigma\Delta\Phi/(JL)$. The relative changes in $F$ through bioturbation are generally less than those in $\kappa$ (Fig. C2).

## Appendix D: Stochastic animal behavior

To account for the stochastic animal behavior in simulations described in Section 3, five runs were performed for each simulation although not all the results are presented in Section 3. We present the five run results for the biodiffusion coefficient in the default simulation in order to demonstrate the stochastic effect on our results (Fig. D1).

## Appendix E: Implicit pumping action

Although eLABS cannot explicitly simulate the pumping action of infauna (e.g., Meysman et al., 2005), a simplified implementation can be made by arbitrarily increasing the constant water flows imposed when organisms ingest/egest sediment particles.  By increasing the advective water flows imposed at the time of ingestion/egestion by factors of $10^2$, $10^3$ and $10^4$ along with otherwise the default settings, advective water flow can mix oxygen more vigorously (e.g., Fig. E1). Note that with the highest enhancement factor, i.e., $10^4$, biology-induced water flow (e.g., ~5 cm$^3$ min$^{-1}$, Fig. E1) becomes comparable to the pumping flow by infauna reported in laboratory studies (e.g., < ~5 cm$^3$ min$^{-1}$, Kristensen, 2001). Corresponding to the deeper oxygen penetration, oxygen consumption by respiration is enhanced and the biodiffusion coefficient becomes large in deep depths (Figs. E2–5).  These results are generally similar to those in the case where shear velocity is high (Section 3.1), suggesting that the effects of increasing advective mixing by ocean current and benthic organisms are similar.  Nonetheless, further model development is necessary to explicitly simulate infaunal pumping action (Section 4).

## Author contributions

YK designed and implemented the model. BPB provided the latest version of LABS code. YK, SKT and AR designed the simulations.  All authors contributed to the writing of the paper.

## Acknowledgments

This research was supported by the Heising-Simons Foundation through a grant to A Ridgwell, S. Kirtland Turner, and L. Kump (#2015-145).  We express our gratitude to two anonymous reviewers for their useful comments and to G. Munhoven for editorial handling of our paper.

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

**Figure captions**

**Figure 1. Illustration of grid on which the behavior of a benthic organism is simulated by the Lattice-Automaton Bioturbation Simulator (LABS). Note that the left and right boundaries of the grid are continuous. Shown in lower right is a magnified view**
**where each grid cell ($0.05 \times 0.05$ cm$^2$) is visible. Dashed lines are drawn to show the implicit width of the system with which the 2D grid system can be converted to a 3D system.**

**Figure 2.** Geometry of burrows after 1-year simulations by LABS with different rules for the preferred direction of organism movement. In (a), the preferred direction is where more labile organic matter exists. In (b), an additional rule is imposed compared to (a), that organisms with greater gut fullness prefer to move in the direction where more water exists.

**Figure 3.** Stream function calculated at 225 model days from the start of a simulation in eLABS using the default settings (Section 3).

**Figure 4.** Time evolution of oxygen profiles calculated for 1-year simulations by eLABS. Six different simulations were conducted (Section 3.1): (a) with the default settings, (b) assuming organisms have tolerance to oxygen-depleted conditions but otherwise default settings, (c) with 10× higher rate for aerobic decomposition of organic matter but otherwise default settings, (d) with 0.6 of porosity but otherwise default settings, (e) with 10× higher shear velocity but otherwise default settings, and (f) without the deterministic water flow calculation with otherwise default settings. Shown are the profiles of oxygen concentration normalized to the constant boundary value ($2.2 \times 10^{-4}$ mol L$^{-1}$) at the top layer of the grid at 25, 50, 100, 150, 200 and 250 model days from the start of the simulations.

**Figure 5.** Time evolution of oxygen fluxes obtained from 1-year simulations with eLABS. See Section 3.1 and caption of Fig. 4 for the details of simulations (a–f). Black dotted curves represent the total oxygen consumption fluxes, orange curves the fluxes of oxygen supply via molecular plus eddy diffusion, and turquoise curves the oxygen consumption fluxes through aerobic degradation of organic matter. Note that the scale of vertical axis is different in (e).

**Figure 6.** Time evolution of burrow geometry obtained from 1-year simulations with eLABS. See Section 3.1 and caption of Fig. 4 for the details of simulations (a–f). Shown are burrow geometries at 25, 50, 100, 150, 200 and 250 model days from the start of the simulations.

**Figure 7.** Time evolution of the biodiffusion coefficient ($D_b$) obtained from 1-year simulations with eLABS. See Section 3.1 and caption of Fig. 4 for the details of simulations (a–f). Note that $D_b$ values are presented only for simulation (a) (denoted as $D_{b,std}$); for other simulations (b–f), the ratios of $D_b$ values to those in (a) ($D_b/D_{b,std}$) are shown to facilitate comparison and dashed vertical lines are references to denote $D_b = D_{b,std}$. Plotted are these values/ratios at 25, 50, 75 … 350 model days from the start of the simulations.

**Figure 8.** Time evolution of oxygen profiles calculated from 10-year simulations with eLABS. Four different simulations were conducted (Section 3.2): (a) with low porosity and no advective water flow, but otherwise default settings, (b) additionally assuming organisms to have tolerance to oxygen-depleted conditions, (c) with 10× higher rate for aerobic decomposition of organic matter but otherwise the same settings as in (a), and (d) with 10× higher sedimentation rate but otherwise the same settings as in (a). Shown are the profiles of oxygen concentration normalized to the constant boundary value ($2.2 \times 10^{-4}$ mol L$^{-1}$) at the top layer of the grid at 25, 50 and 100 model days and 1, 5 and 10 model years from the start of the simulations.

**Figure 9.** Time evolution of oxygen fluxes obtained from 10-year simulations with eLABS. See Section 3.2 and caption of Fig. 8 for the details of simulations (a–d). Black dotted curves represent the total oxygen consumption fluxes, orange curves the fluxes of oxygen supply via molecular plus eddy diffusion, and turquoise curves the oxygen consumption fluxes through aerobic degradation of organic matter.

**Figure 10.** Time evolution of burrow geometry obtained from 10-year simulations with eLABS. See Section 3.2 and caption of Fig. 8 for the details of simulations (a–d). Shown are burrow geometries at 25, 50 and 100 model days and 1, 5 and 10 model years from the start of the simulations.

**Figure 11.** Time evolution of the biodiffusion coefficient ($D_b$) obtained from 10-year simulations with eLABS. See Section 3.2 and caption of Fig. 8 for the details of simulations (a–d). Note that the $D_b$ values are presented only for simulation (a) (denoted as $D_{b,std}$); for other simulations (b–d), the ratios of $D_b$ values to those in (a) ($D_b/D_{b,std}$) are shown to facilitate comparison and dashed vertical lines are references to denote $D_b = D_{b,std}$. Plotted are these values/ratios at 25 model days to 10 model years from the start of the simulations with intervals of 25 model days.

**Figure 12.** Time evolution of oxygen profiles calculated from 10-year simulations with eLABS. Four different simulations were conducted (Section 3.3): (a) with low porosity, no advective water flow and 10× lower oxygen concentration for seawater, but otherwise default settings, (b) additionally assuming organisms to have tolerance to oxygen-depleted conditions, (c) with 10× higher rate for aerobic decomposition of organic matter but otherwise the same settings as in (a), and (d) with 10× higher sedimentation rate but otherwise the same settings as in (a). Shown are the profiles of oxygen concentration normalized to the constant boundary value ($2.2 \times 10^{-5}$ mol L$^{-1}$) at the top layer of the grid at 25, 50 and 100 model days and 1, 5 and 10 model years from the start of the simulations.

**Figure 13.** Time evolution of oxygen fluxes obtained from 10-year simulations with eLABS. See Section 3.3 and caption of Fig. 12 for the details of simulations (a–d). Black dotted curves represent the total oxygen consumption fluxes, orange curves the fluxes of oxygen supply via molecular plus eddy diffusion, and turquoise curves the oxygen consumption fluxes through aerobic degradation of organic matter.

**Figure 14.** Time evolution of burrow geometry obtained from 10-year simulations with eLABS. See Section 3.3 and caption of Fig. 12 for the details of simulations (a–d). Shown are burrow geometries at 25, 50 and 100 model days and 1, 5 and 10 model years from the start of the simulations.

**Figure 15.** Time evolution of the biodiffusion coefficient ($D_b$) obtained from 10-year simulations with eLABS. See Section 3.3 and caption of Fig. 12 for the details of simulations (a–d). Note that the $D_b$ values are presented only for simulation (a) (denoted as $D_{b,std}$); for other simulations (b–d), the ratios of $D_b$ values to those in (a) ($D_b/D_{b,std}$) are shown to facilitate comparison and dashed vertical lines are references to denote $D_b = D_{b,std}$. Plotted are these values/ratios at 25 model days to 10 model years from the start of the simulations with intervals of 25 model days.

**Figure A1.** All oxygen fluxes calculated in eLABS. Shown are oxygen fluxes caused by a change in the total amount of oxygen in the sediment ('Non-steady-state'), aerobic degradation of organic matter ('Aerobic OM decomposition'), infaunal respiration ('Respiration'), water advection ('Advective supply') and molecular plus eddy diffusion ('Diffusive supply'), and also the residual oxygen flux as the sum of the above fluxes ('Residual') in (a) default simulation and (c) simulation without oxygen consumption and ingestion/egestion. Fig. A1b shows only residual oxygen flux in a.

**Figure B1.** Time evolution of oxygen profiles in simulations (a) with the default settings, (b) assuming a higher resolution (480×480) for the water flow calculation with otherwise default settings and (c) assuming a wider calculation domain (24×12 cm$^2$) and two deposit feeders with otherwise default settings.

**Figure B2.** Time evolution of oxygen fluxes in simulations (a) with the default settings, (b) assuming a higher resolution (480×480) for the water flow calculation with otherwise default settings and (c) assuming a wider calculation domain (24×12 cm$^2$) and two deposit feeders with otherwise default settings. Only the absolute flux values of diffusive oxygen supply, aerobic decomposition of OM and total oxygen consumption are shown.

**Figure B3.** Time evolution of burrow geometry in simulations (a) with the default settings, (b) assuming a higher resolution (480×480) for the water flow calculation with otherwise default settings and (c) assuming a wider calculation domain (24×12 cm$^2$) and two deposit feeders with otherwise default settings.

**Figure B4.** Time evolution of the biodiffusion coefficient ($D_b$) in simulations (a) with the default settings, (b) assuming a higher resolution (480×480) for the water flow calculation with otherwise default settings and (c) assuming a wider calculation domain (24×12 cm$^2$) and two deposit feeders with otherwise default settings. Note that the $D_b$ values are presented only for simulation (a) (denoted as $D_{b,std}$); for other simulations (b and c), the ratios of $D_b$ values to those in (a) ($D_b/D_{b,std}$) are shown to facilitate comparison and dashed vertical lines are references to denote $D_b = D_{b,std}$.

**Figure C1.** Streamlines for permeability calculation at the beginning of simulation (a) in Section 3.2.

**Figure C2.** Relative changes of permeability and formation factor in the default simulation (simulation (a) in Section 3.1).

**Figure C3.** Electric potentials for formation factor calculation at the beginning of simulation (a) in Section 3.2.

**Figure D1.** Time evolution of the biodiffusion coefficient ($D_b$) for 5 runs of default simulation (simulation (a) in Section 3.1) in order to illustrate the effect of stochastic animal behavior.

**Figure E1.** Stream function calculated at 175 model days in simulation with water flows increased at the time of ingestion/egestion by a factor of 10$^4$.

**Figure E2.** Time evolution of oxygen profiles in simulations (a) with 10× higher shear velocity but otherwise default settings and (b–d) increasing water flows at the time of ingestion/egestion by factors of (b) 10$^2$, (c) 10$^3$ and (d) 10$^4$ but with otherwise default settings.

**Figure E3.** Time evolution of oxygen fluxes in simulations (a) with 10× higher shear velocity but otherwise default settings and (b–d) increasing water flows at the time of ingestion/egestion by factors of (b) 10$^2$, (c) 10$^3$ and (d) 10$^4$ but with otherwise default settings. Only the absolute flux values of diffusive oxygen supply, aerobic decomposition of OM and total oxygen consumption are shown.

**Figure E4.** Time evolution of burrow geometry in simulations (a) with 10× higher shear velocity but otherwise default settings and (b–d) increasing water flows at the time of ingestion/egestion by factors of (b) 10$^2$, (c) 10$^3$ and (d) 10$^4$ but with otherwise default settings.

**Figure E5.** Time evolution of the biodiffusion coefficient ($D_b$) in simulations (a) with 10× higher shear velocity but otherwise default settings and (b–d) increasing water flows at the time of ingestion/egestion by factors of (b) $10^2$, (c) $10^3$ and (d) $10^4$ but with otherwise default settings. Note that the ratios of $D_b$ values to those in default simulation ($D_{b,std}$ in Fig. 7a) ($D_b/D_{b,std}$) are shown to facilitate comparison and dashed vertical lines are references to denote $D_b = D_{b,std}$.

Table 1. Model parameterization.

| Parameter [a] | Section 3.1 (Figs. 4–7) | | | | | | Section 3.2 (Figs. 8–11) | | | | Section 3.3 (Figs. 12–15) | | | |
|---|---|---|---|---|---|---|---|---|---|---|---|---|---|---|
| | (a) | (b) | (c) | (d) | (e) | (f) | (a) | (b) | (c) | (d) | (a) | (b) | (c) | (d) |
| Tolerance to hypoxia [b] | F | T | F | F | F | F | F | T | F | F | F | T | F | F |
| OM decomposition rate ($10^2$ wt%$^{-1}$ yr$^{-1}$) | 4.54 | 4.54 | 45.4 | 4.54 | 4.54 | 4.54 | 4.54 | 4.54 | 45.4 | 4.54 | 4.54 | 4.54 | 45.4 | 4.54 |
| Porosity | 0.8 | 0.8 | 0.8 | 0.6 | 0.8 | 0.8 | 0.6 | 0.6 | 0.6 | 0.6 | 0.6 | 0.6 | 0.6 | 0.6 |
| Shear velocity ($10^6$ cm yr$^{-1}$) | 1 | 1 | 1 | 1 | 10 | 1 | 1 | 1 | 1 | 1 | 1 | 1 | 1 | 1 |
| Advective flow [b] | T | T | T | T | T | F | F | F | F | F | F | F | F | F |
| Sedimentation rate ($10^{-2}$ cm yr$^{-1}$) | 1.5 | 1.5 | 1.5 | 1.5 | 1.5 | 1.5 | 1.5 | 1.5 | 1.5 | 15 | 1.5 | 1.5 | 1.5 | 15 |
| Oxygen concentration ($10^{-4}$ mol L$^{-1}$) | 2.2 | 2.2 | 2.2 | 2.2 | 2.2 | 2.2 | 2.2 | 2.2 | 2.2 | 2.2 | 0.22 | 0.22 | 0.22 | 0.22 |
| Simulation duration (yr) | 1 | 1 | 1 | 1 | 1 | 1 | 10 | 10 | 10 | 10 | 10 | 10 | 10 | 10 |

[a] See Sections 2 and 3 for the details.

[b] 'F' denotes that simulation either assumes no tolerance to hypoxia for organisms or does not account for advective water flow caused by animal movements. 'T' denotes the opposite parameterization to 'F'.

Figure 1

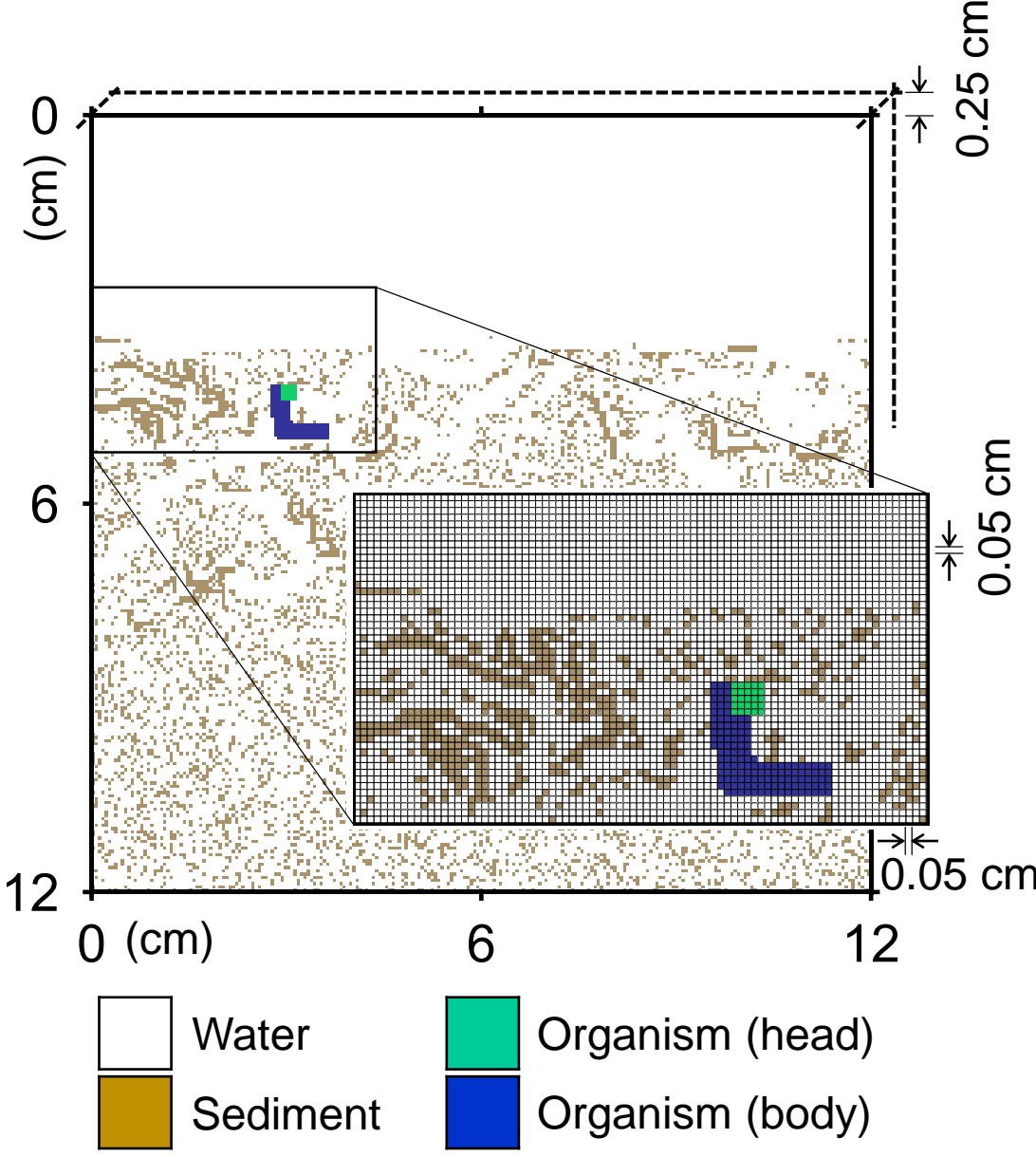

Water · · · · · · Organism (head)

Sediment · · · · Organism (body)

Figure 2

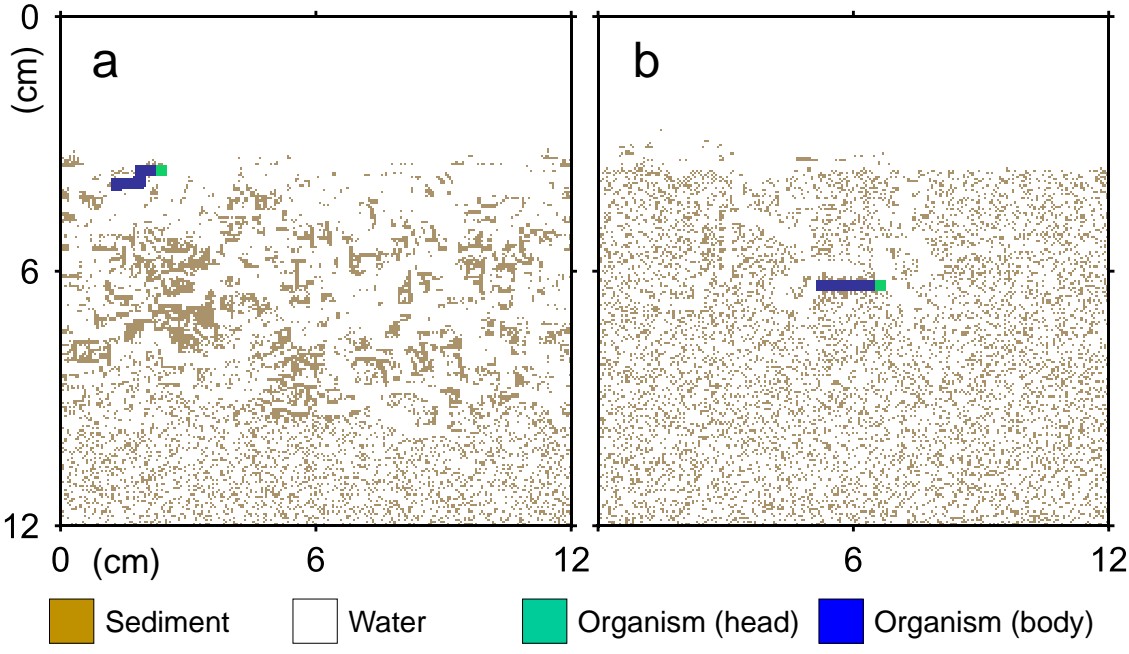

Figure 3

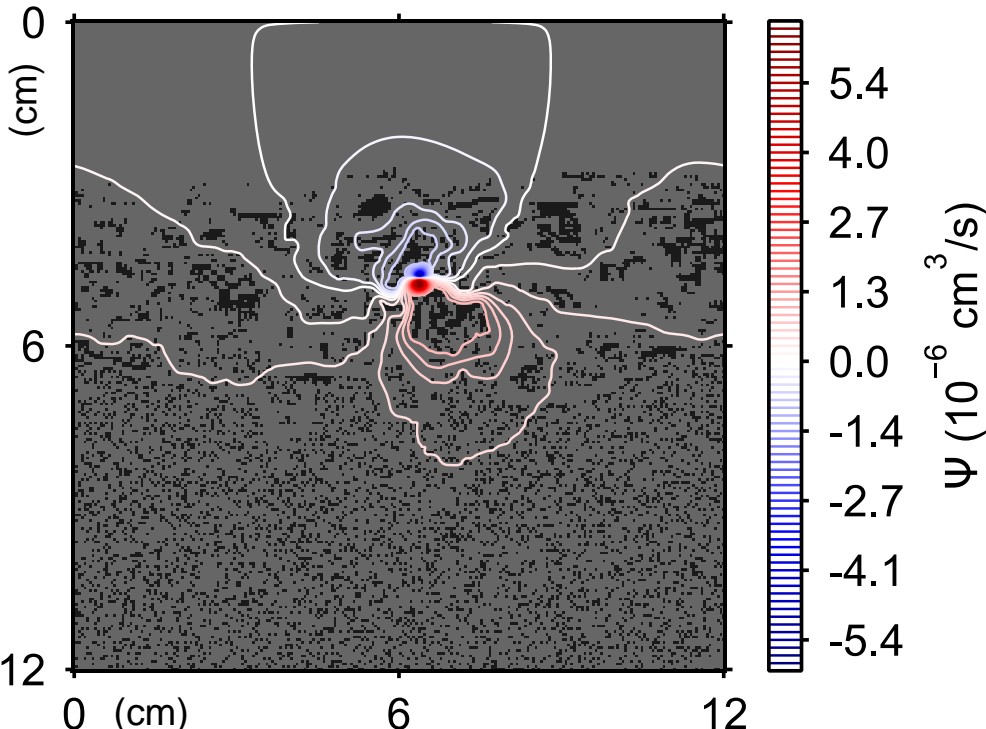

Figure 4

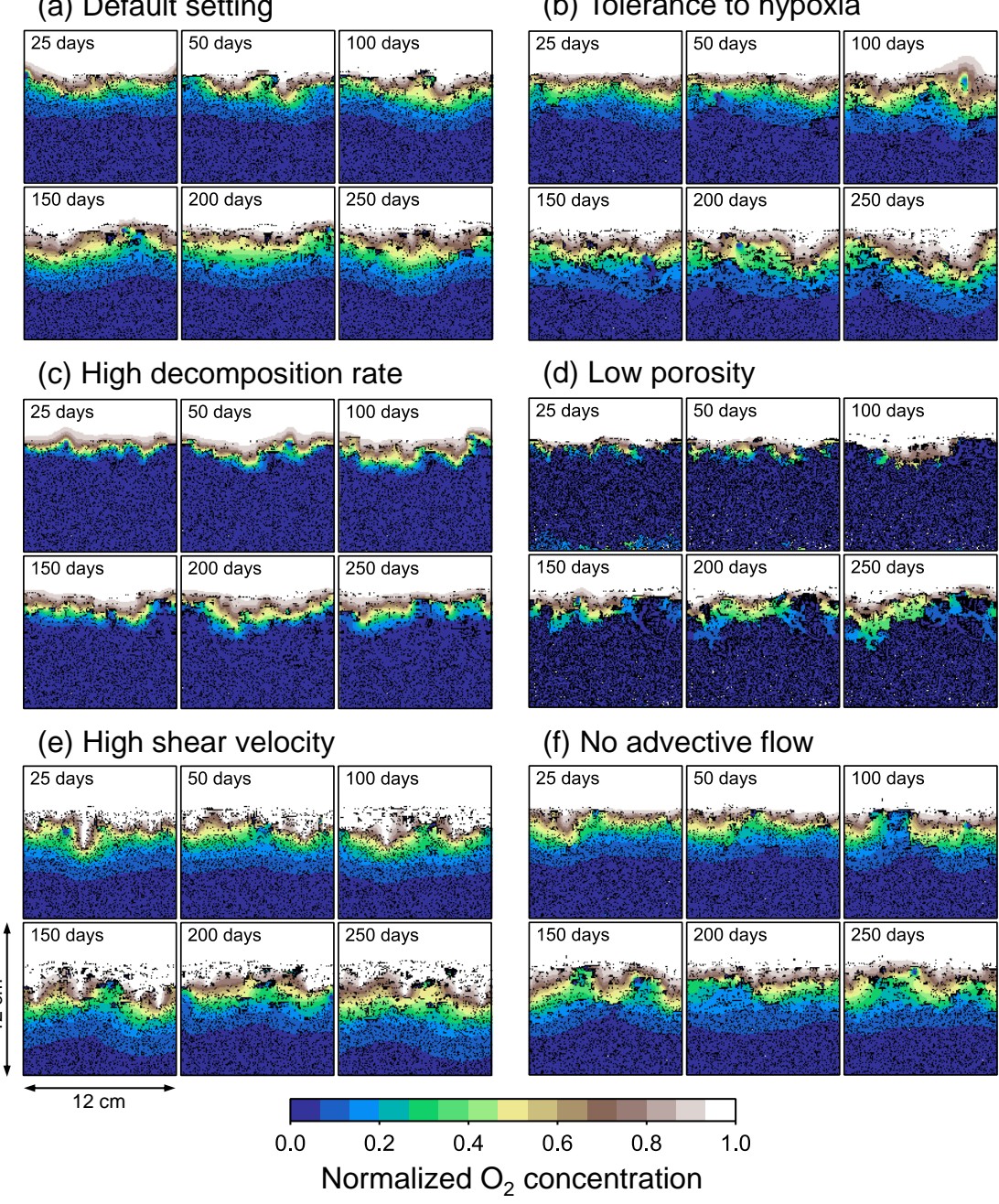

(a) Default setting

(b) Tolerance to hypoxia

(c) High decomposition rate

(d) Low porosity

(e) High shear velocity

(f) No advective flow

0.0   0.2   0.4   0.6   0.8   1.0
Normalized O$_2$ concentration

Figure 5

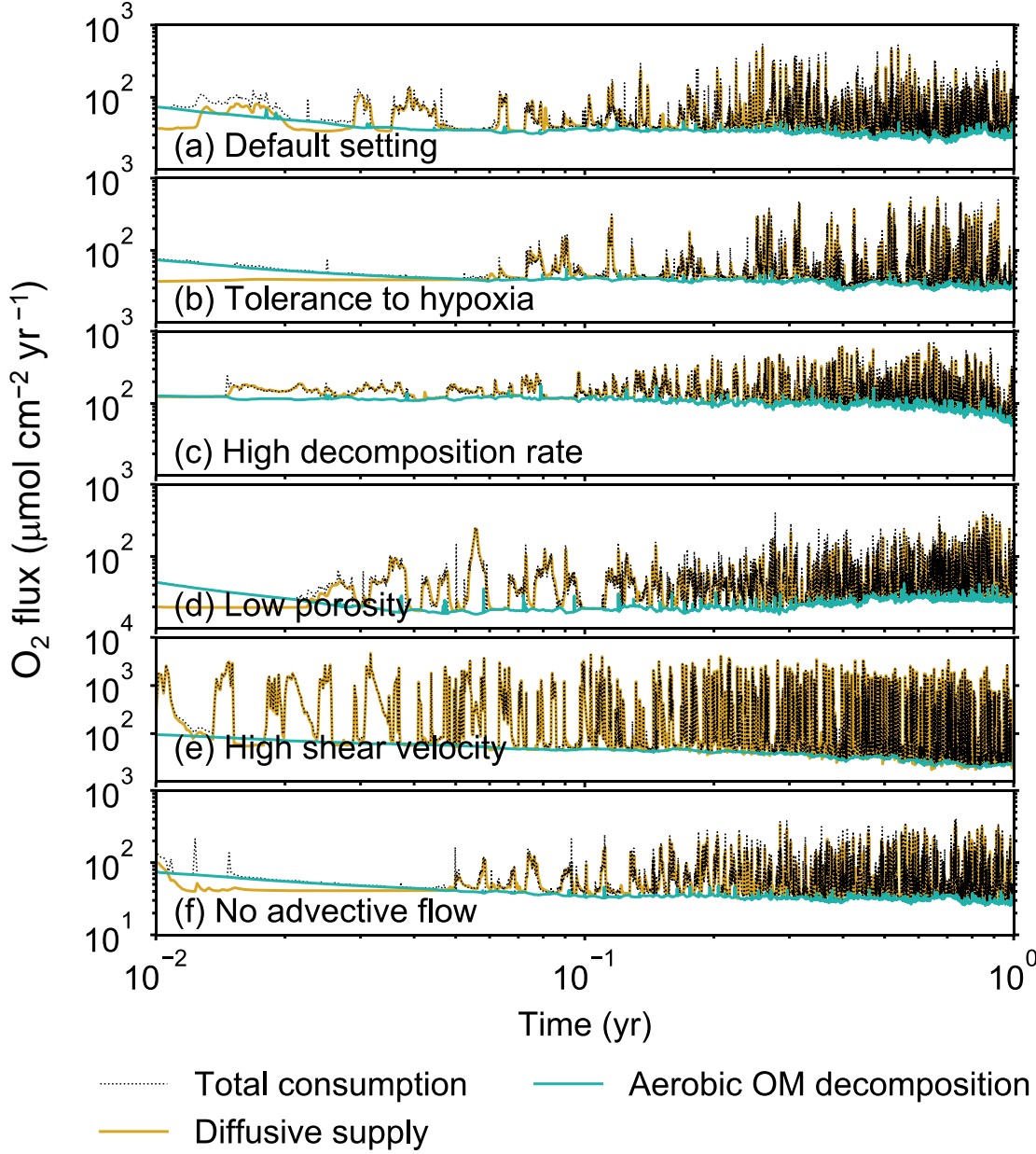

Figure 6

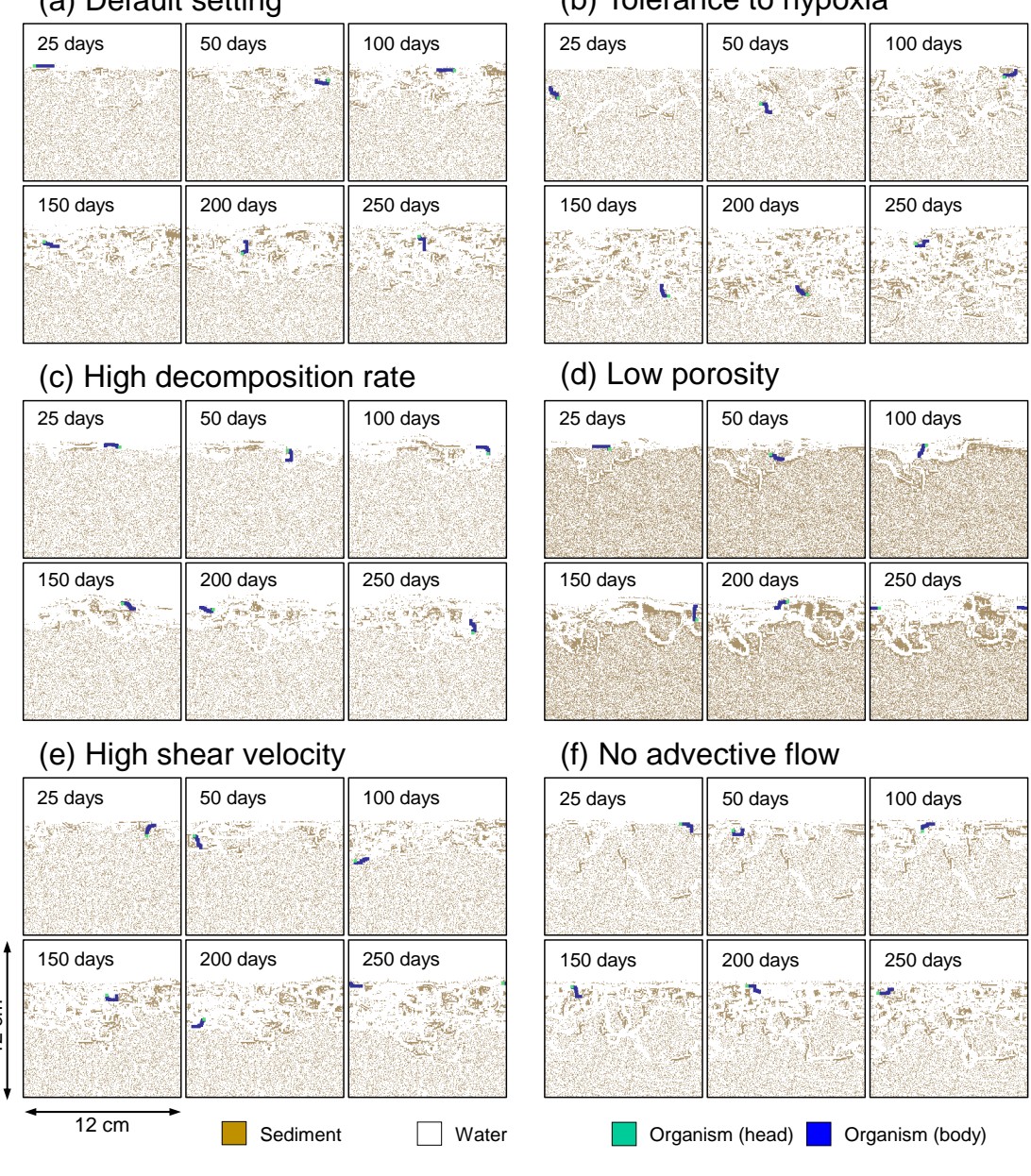

Figure 7

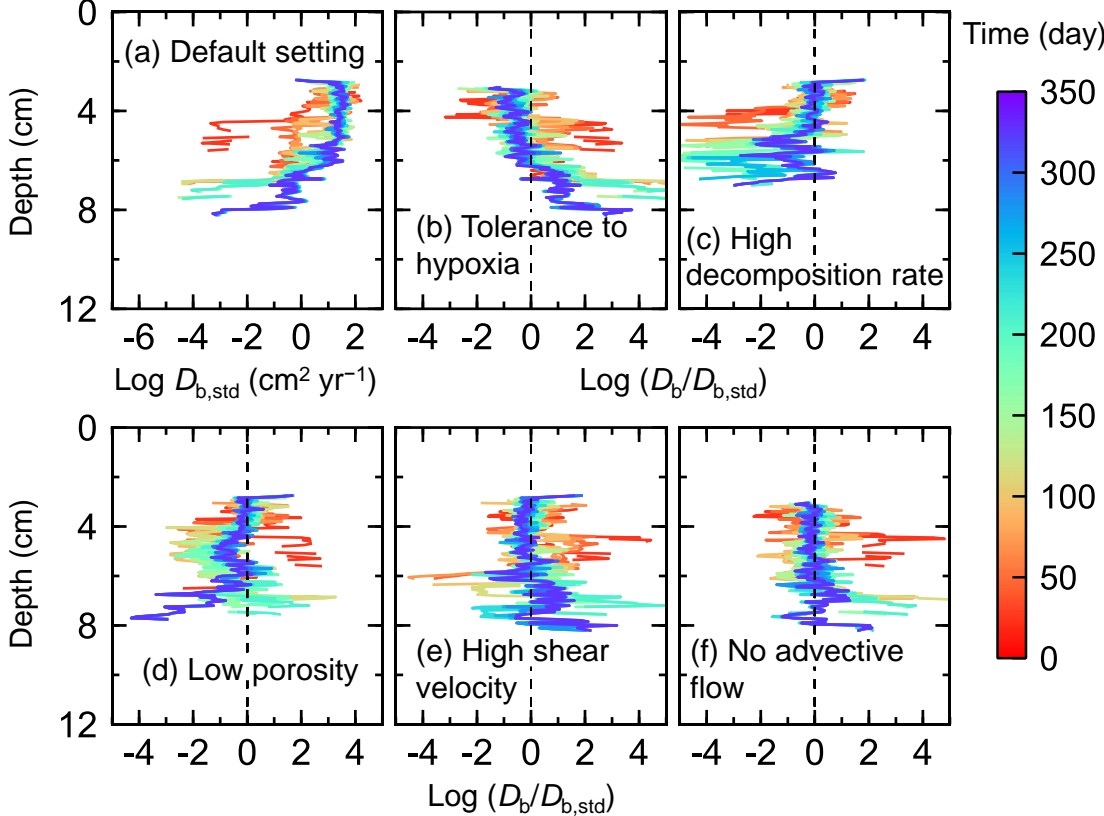

Figure 8

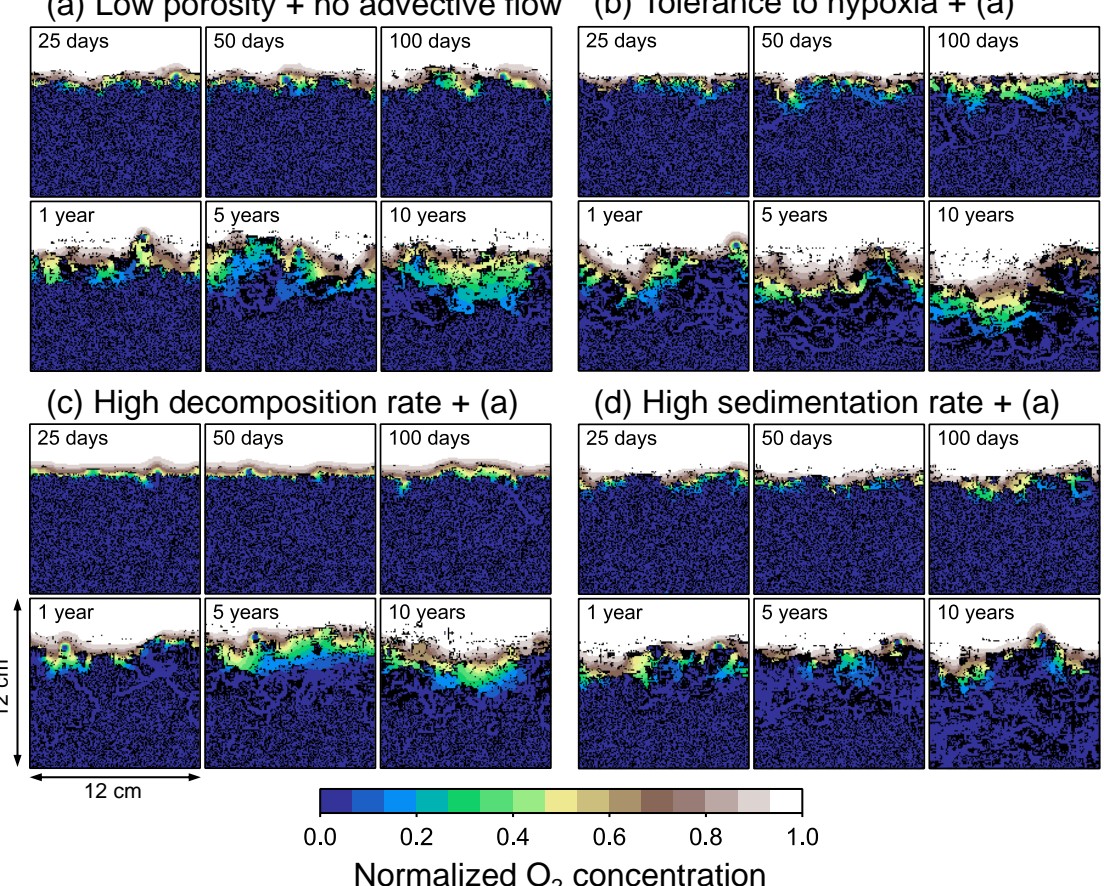

(a) Low porosity + no advective flow

(b) Tolerance to hypoxia + (a)

(c) High decomposition rate + (a)

(d) High sedimentation rate + (a)

25 days    50 days    100 days

1 year    5 years    10 years

12 cm

12 cm

Normalized O$_2$ concentration

0.0    0.2    0.4    0.6    0.8    1.0

Figure 9

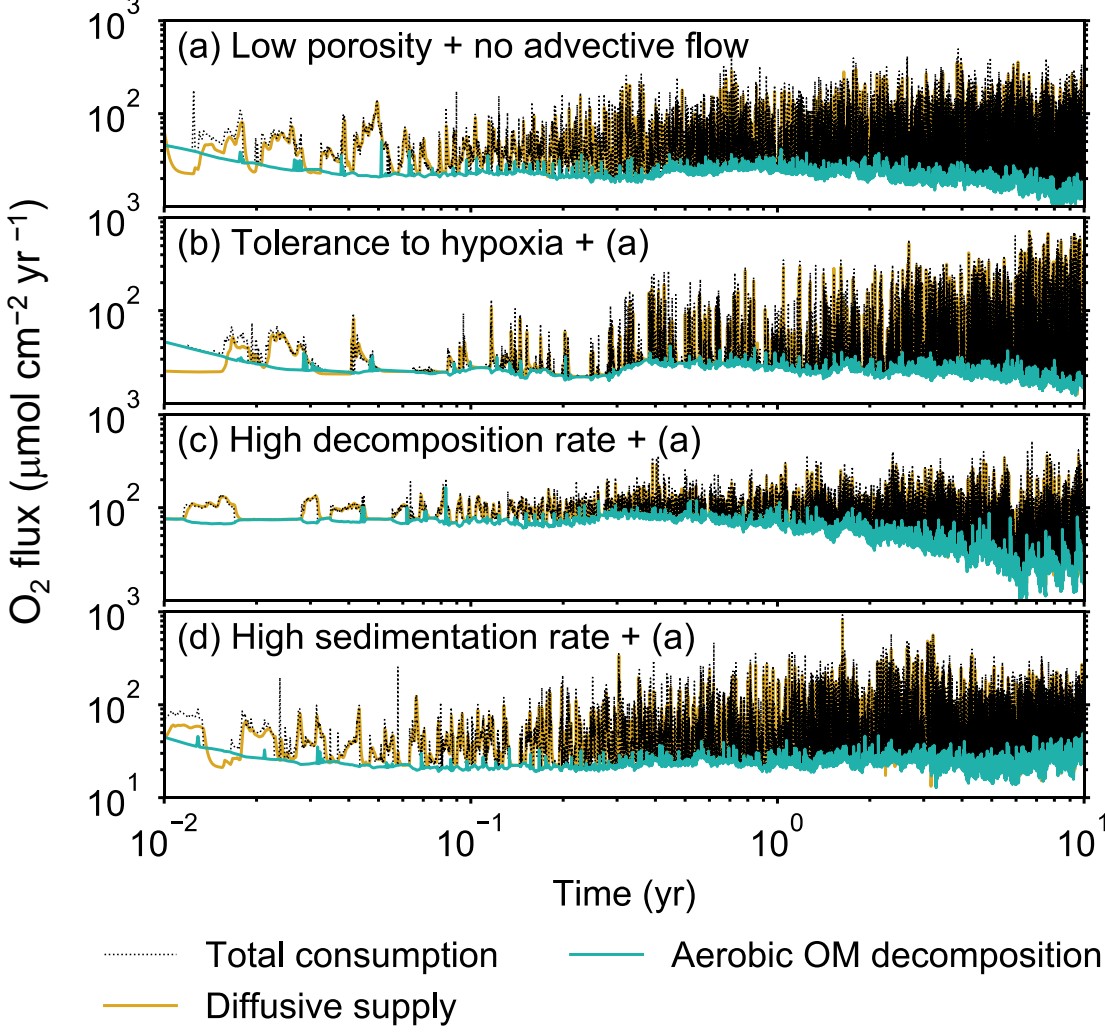

Figure 10

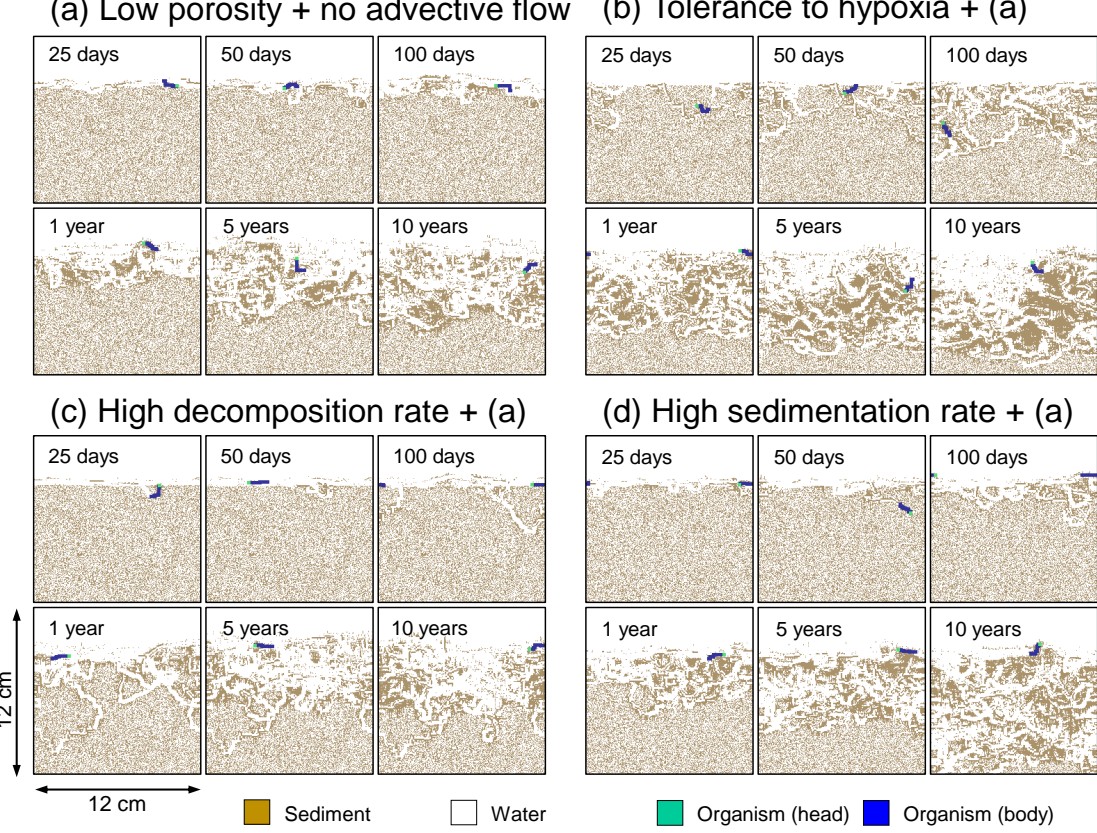

(a) Low porosity + no advective flow

25 days  50 days  100 days
1 year  5 years  10 years

(b) Tolerance to hypoxia + (a)

25 days  50 days  100 days
1 year  5 years  10 years

(c) High decomposition rate + (a)

25 days  50 days  100 days
1 year  5 years  10 years

(d) High sedimentation rate + (a)

25 days  50 days  100 days
1 year  5 years  10 years

12 cm

12 cm

Sediment    Water    Organism (head)    Organism (body)

Figure 11

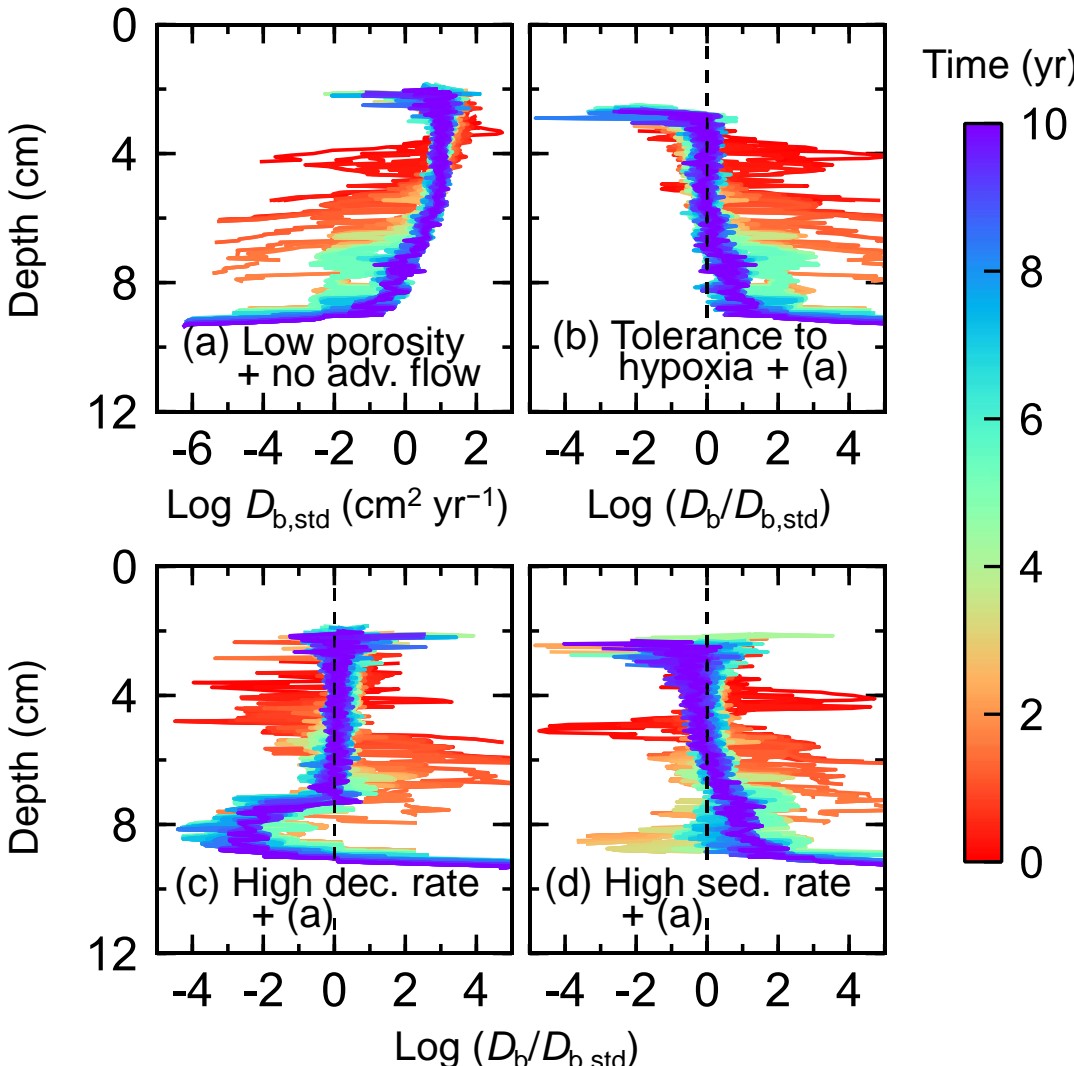

Figure 12

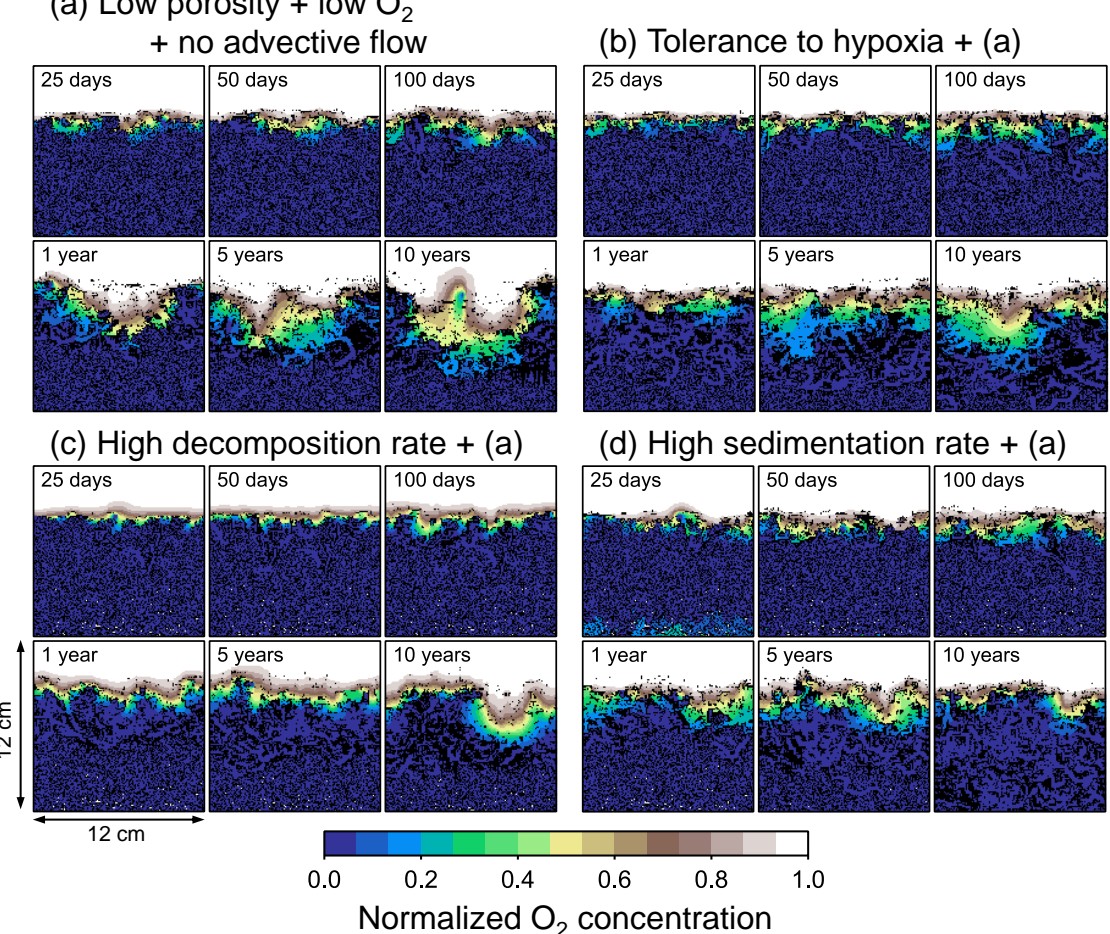

(a) Low porosity + low O$_2$ + no advective flow

(b) Tolerance to hypoxia + (a)

(c) High decomposition rate + (a)

(d) High sedimentation rate + (a)

25 days  50 days  100 days  1 year  5 years  10 years

12 cm

12 cm

0.0  0.2  0.4  0.6  0.8  1.0

Normalized O$_2$ concentration

Figure 13

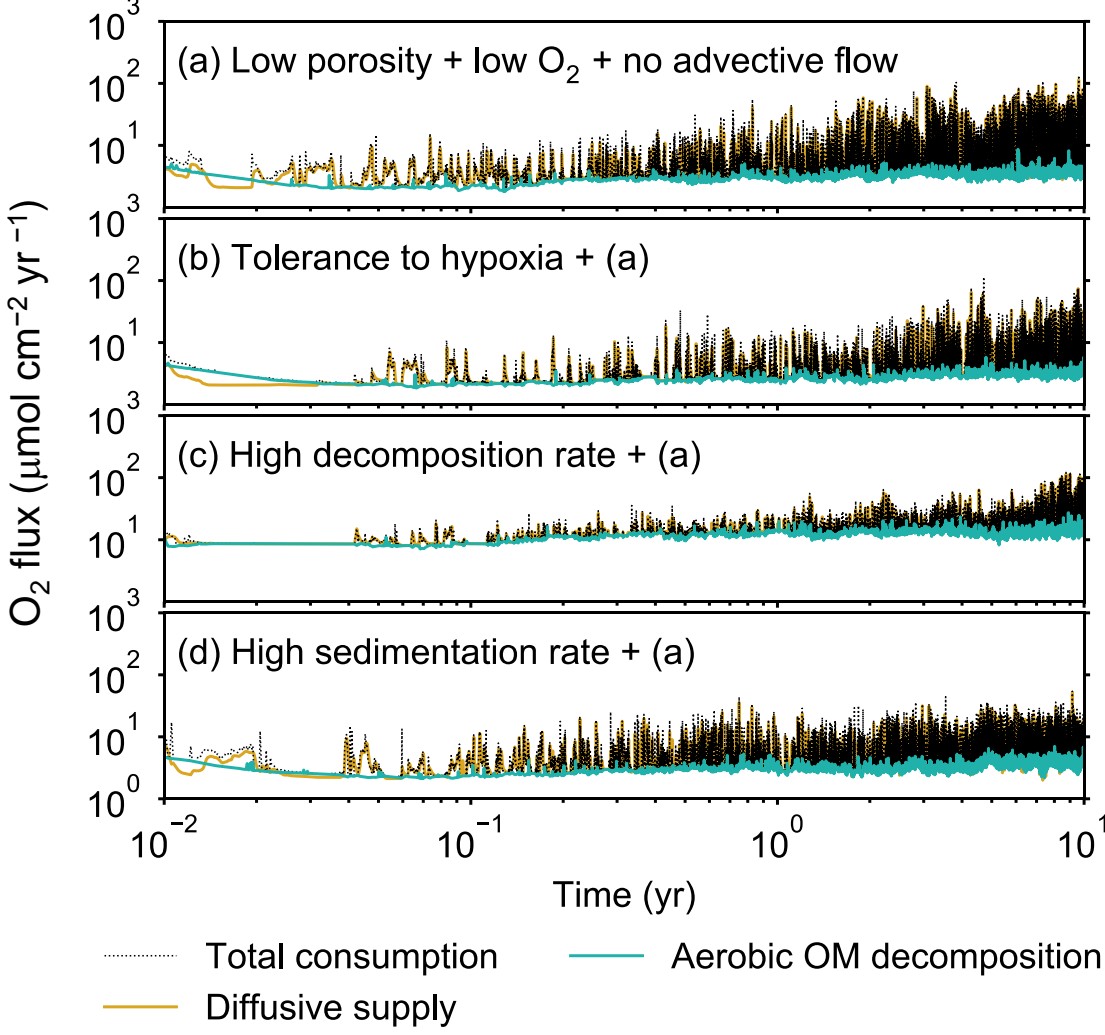

O$_2$ flux ($\mu$mol cm$^{-2}$ yr$^{-1}$)

(a) Low porosity + low O$_2$ + no advective flow

(b) Tolerance to hypoxia + (a)

(c) High decomposition rate + (a)

(d) High sedimentation rate + (a)

Time (yr)

............ Total consumption  —— Aerobic OM decomposition

—— Diffusive supply

Figure 14

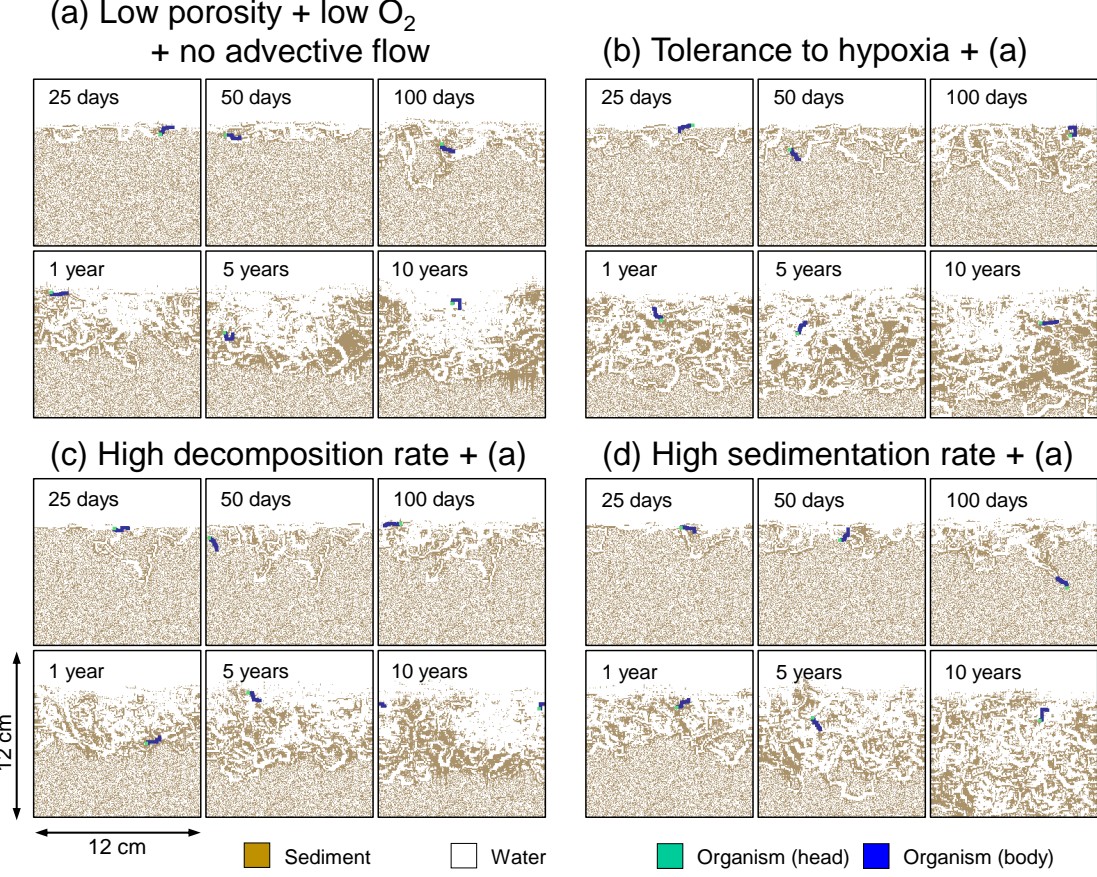

(a) Low porosity + low O$_2$ + no advective flow

(b) Tolerance to hypoxia + (a)

(c) High decomposition rate + (a)

(d) High sedimentation rate + (a)

25 days   50 days   100 days
1 year    5 years   10 years

12 cm

12 cm

■ Sediment   □ Water   ■ Organism (head)   ■ Organism (body)

Figure 15

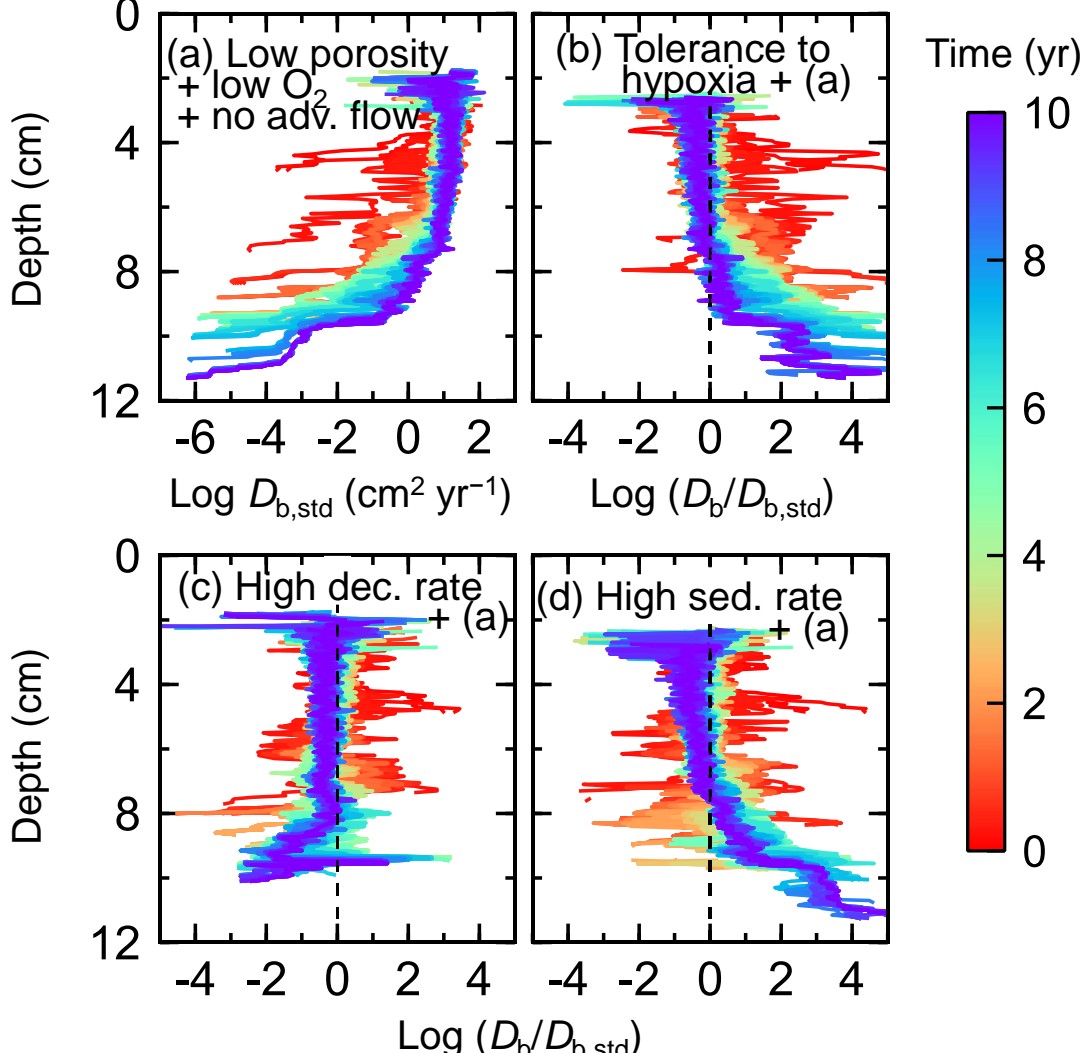

Figure A1

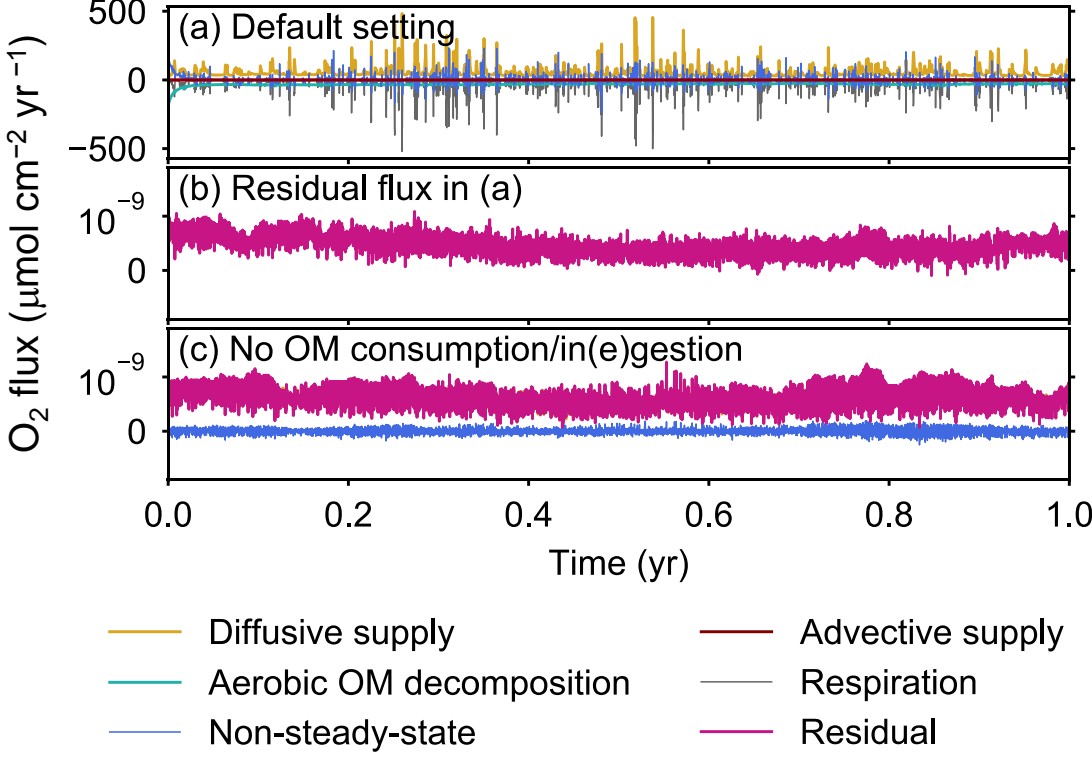

Figure B1

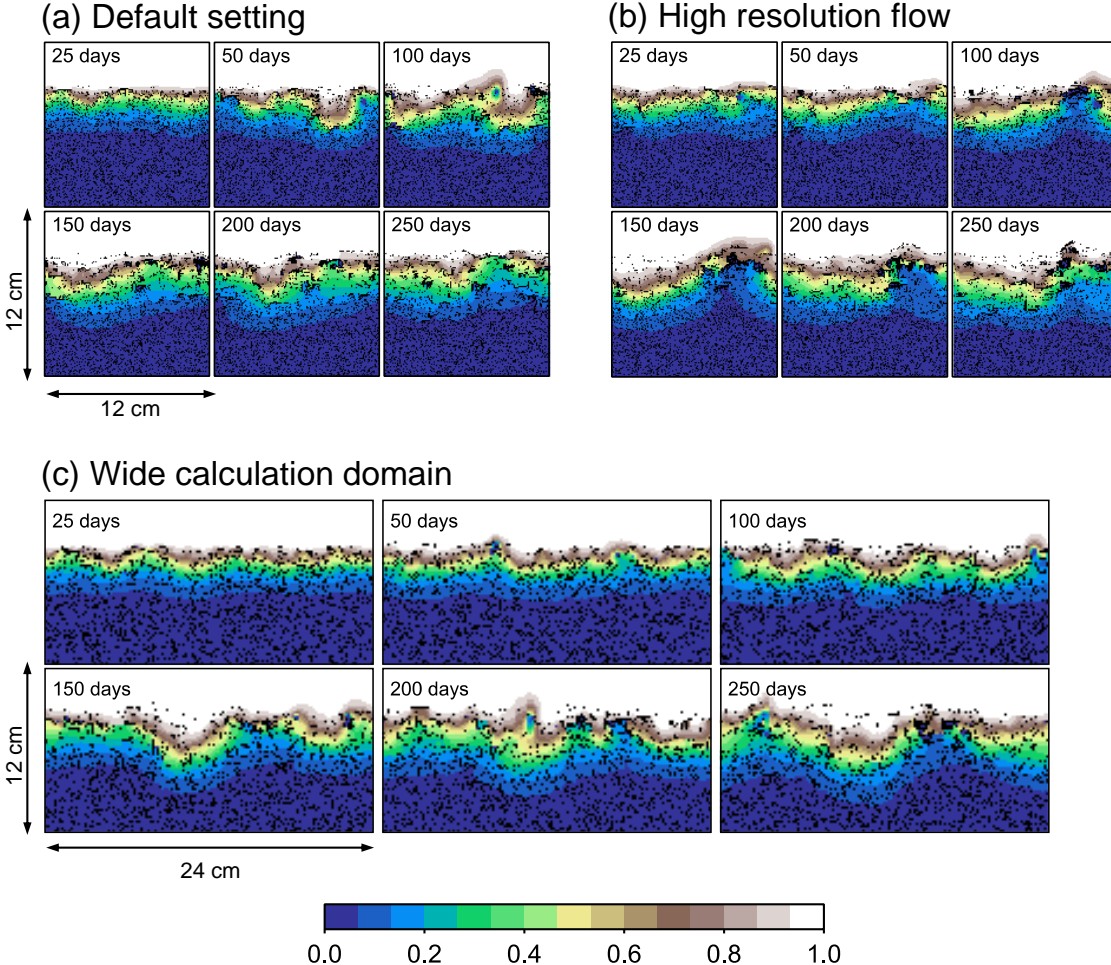

(a) Default setting

25 days    50 days    100 days

150 days    200 days    250 days

12 cm

12 cm

(b) High resolution flow

25 days    50 days    100 days

150 days    200 days    250 days

(c) Wide calculation domain

25 days    50 days    100 days

150 days    200 days    250 days

12 cm

24 cm

0.0    0.2    0.4    0.6    0.8    1.0

Normalized O$_2$ concentration

Figure B2

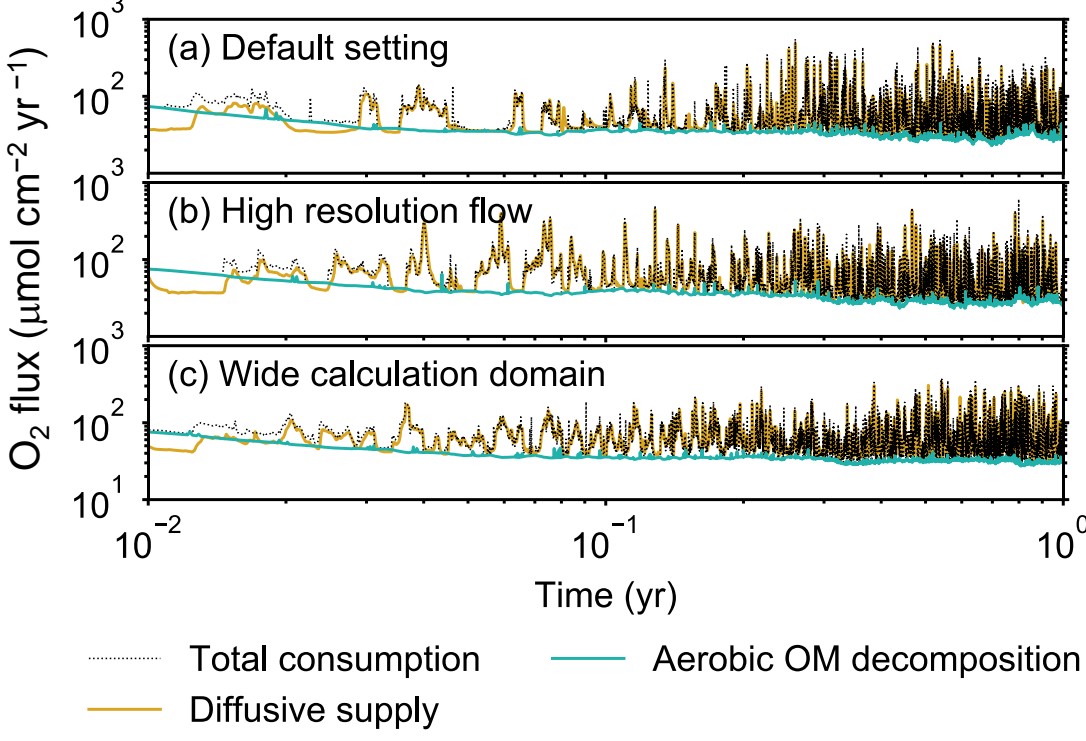

Figure B3

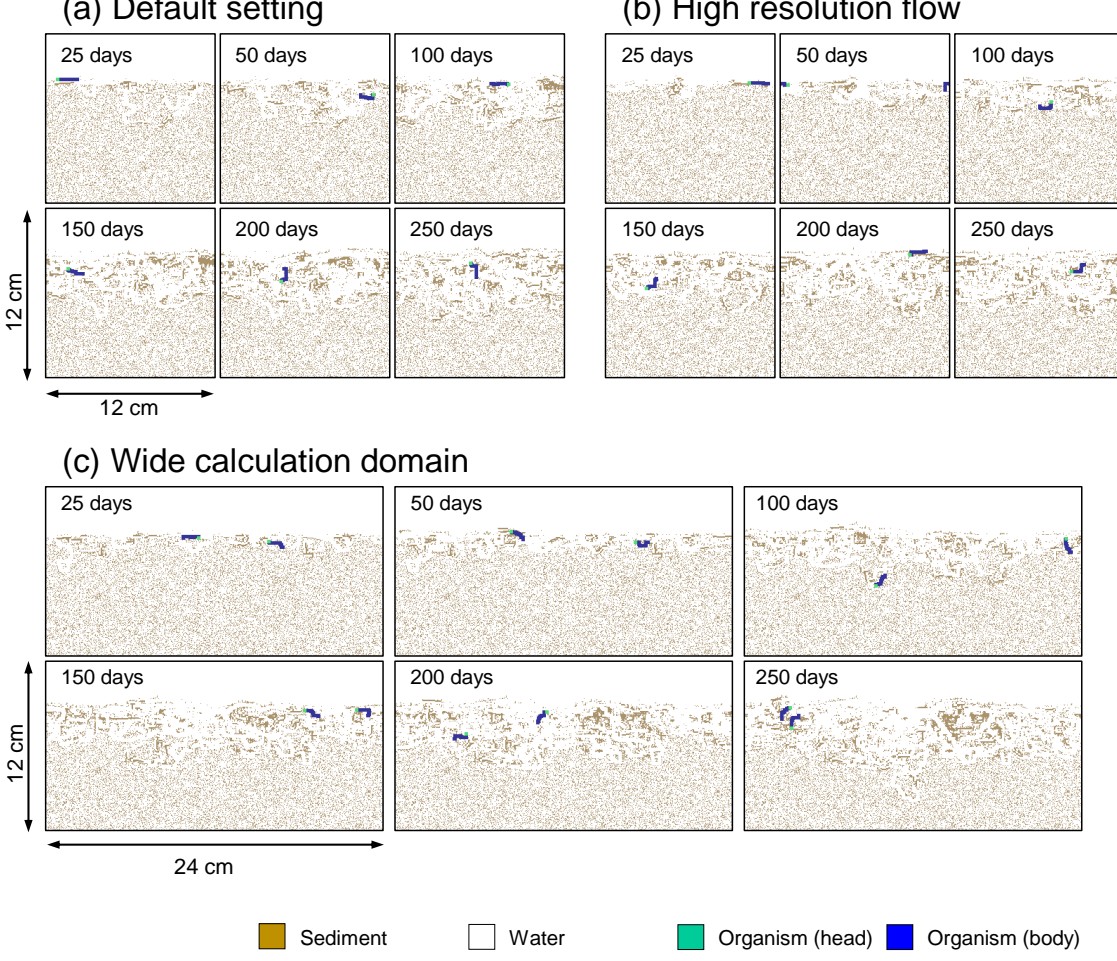

(a) Default setting

(b) High resolution flow

(c) Wide calculation domain

Sediment   Water   Organism (head)   Organism (body)

Figure B4

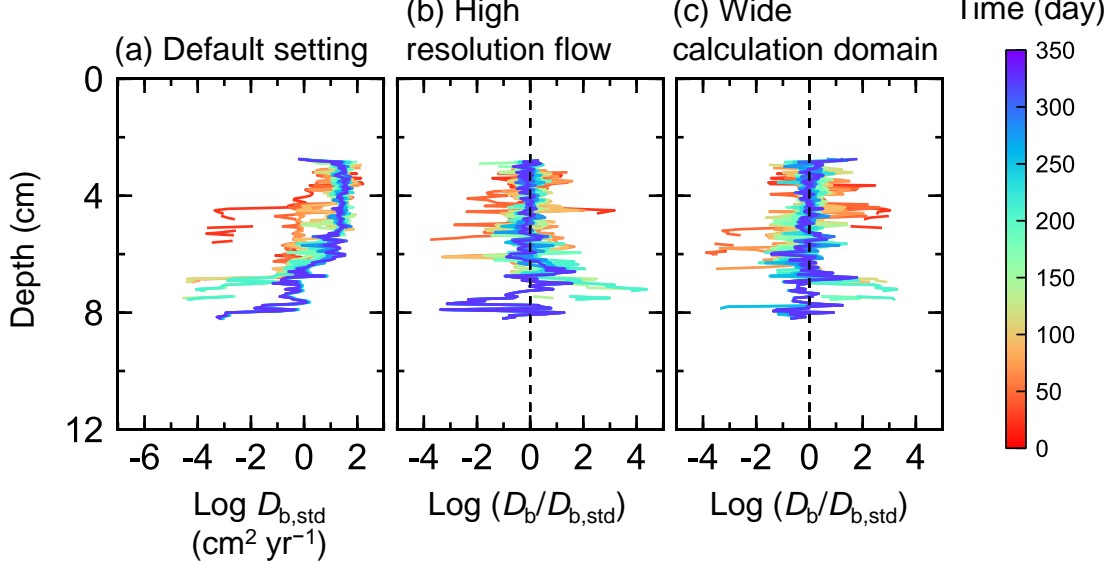

Figure C1

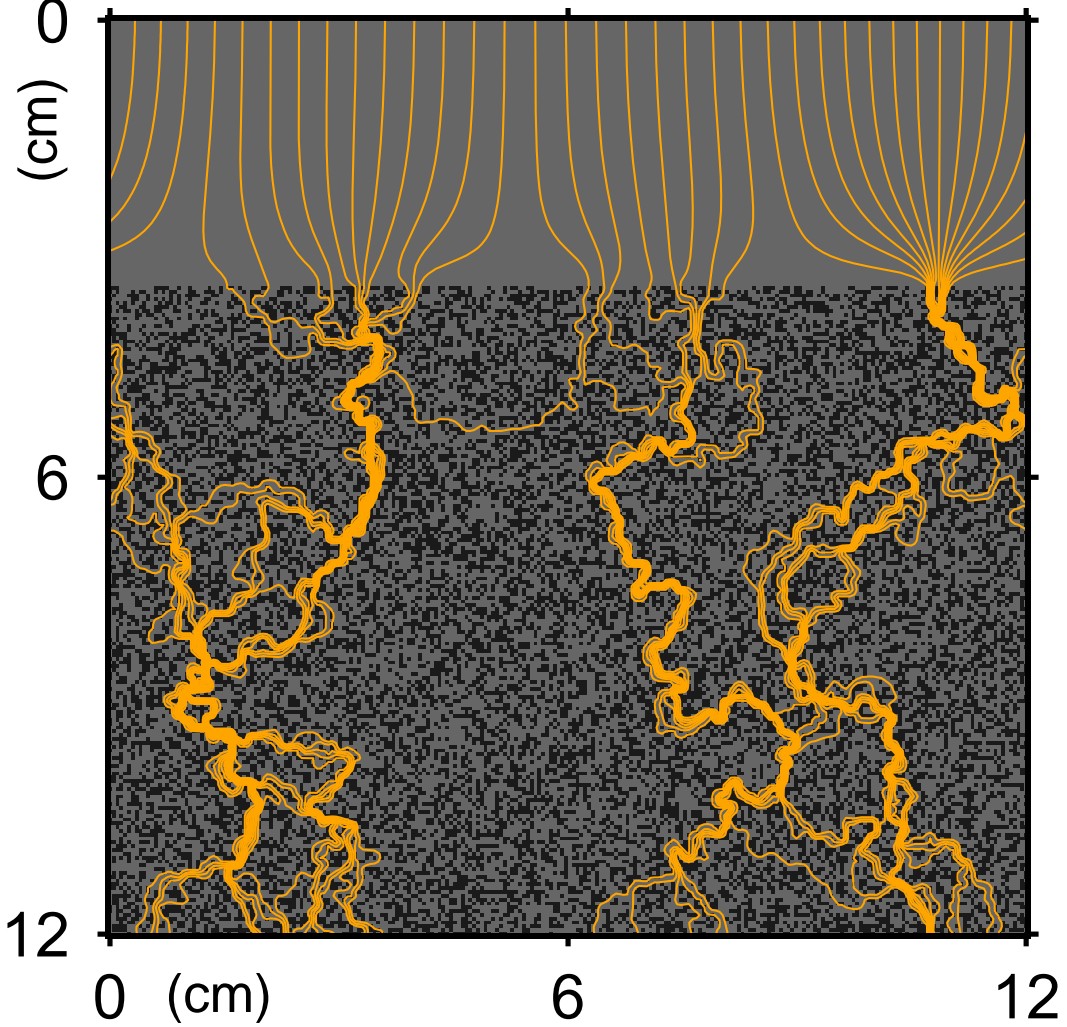

Figure C2

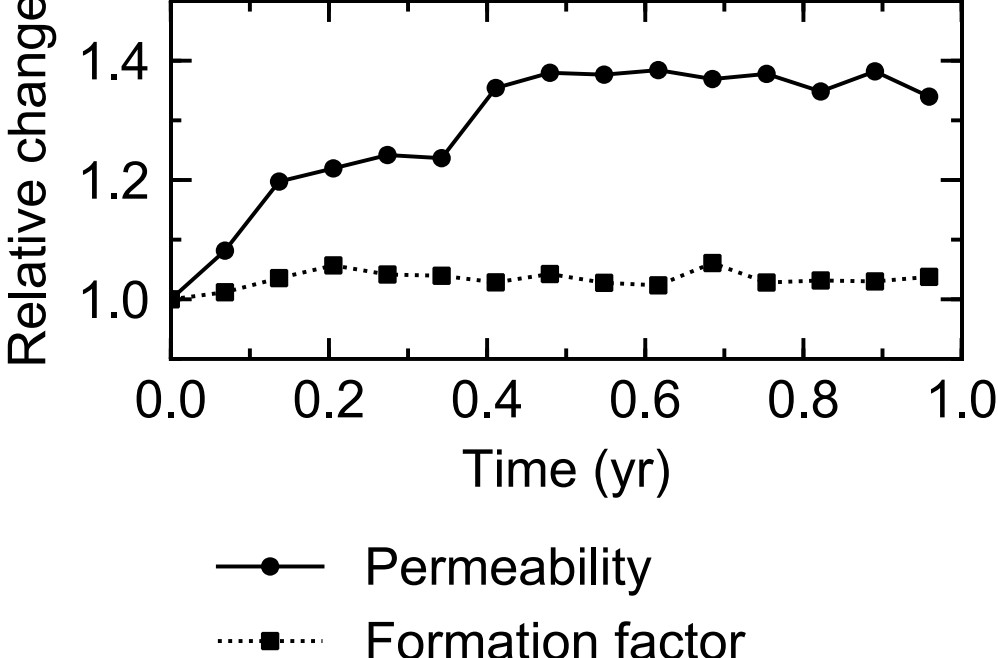

Figure C3

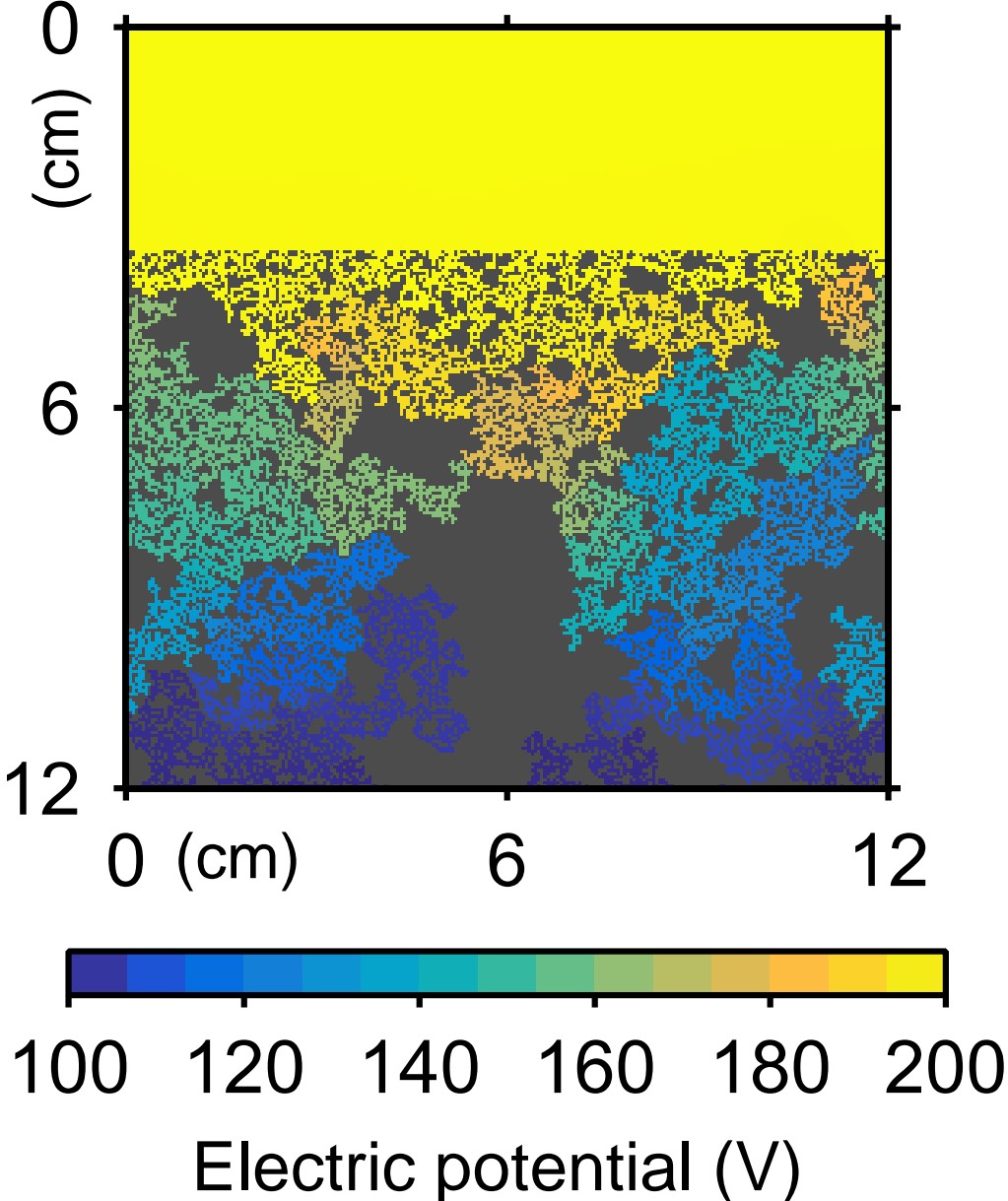

Electric potential (V)

Figure D1

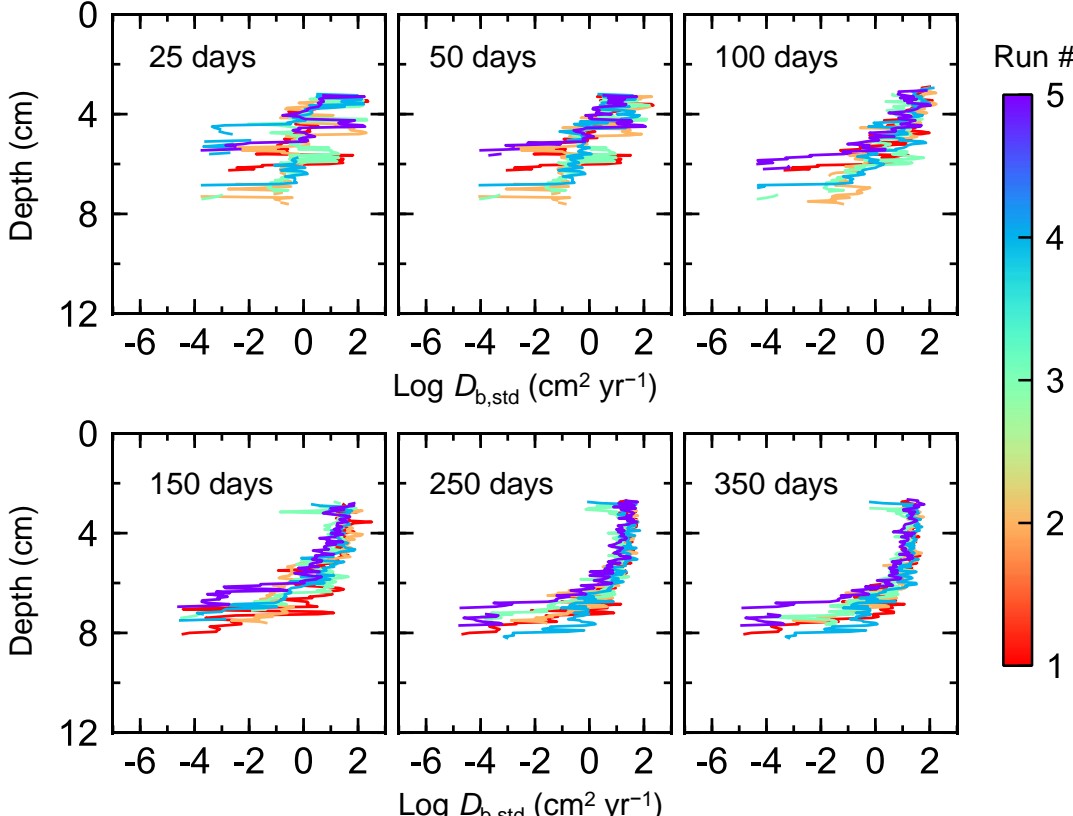

Figure E1

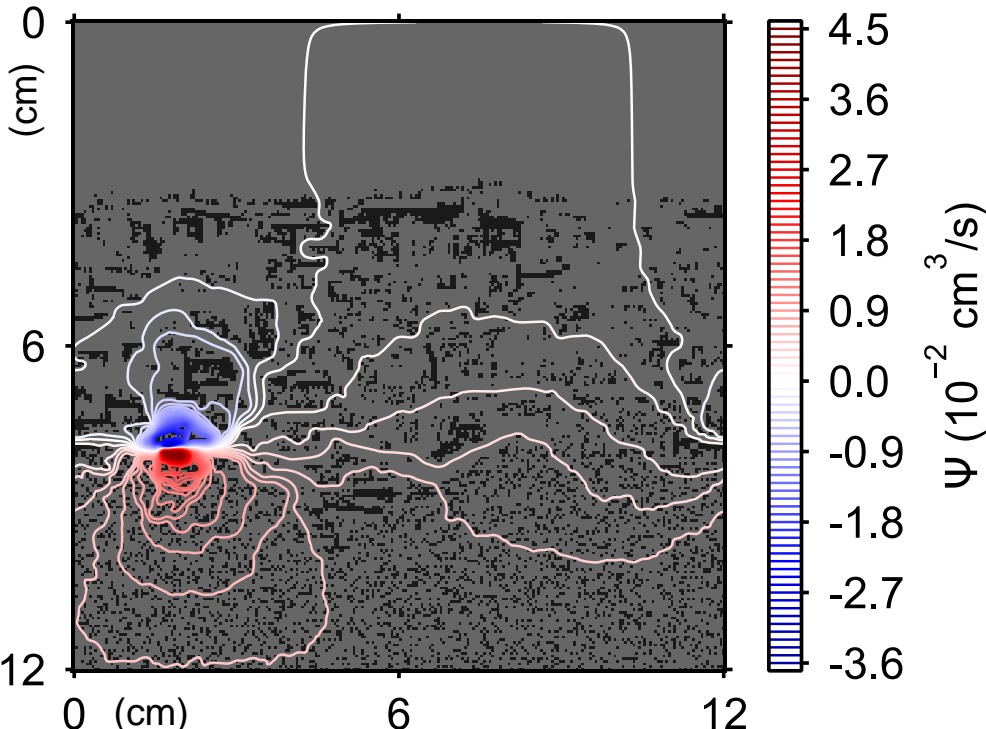

Figure E2

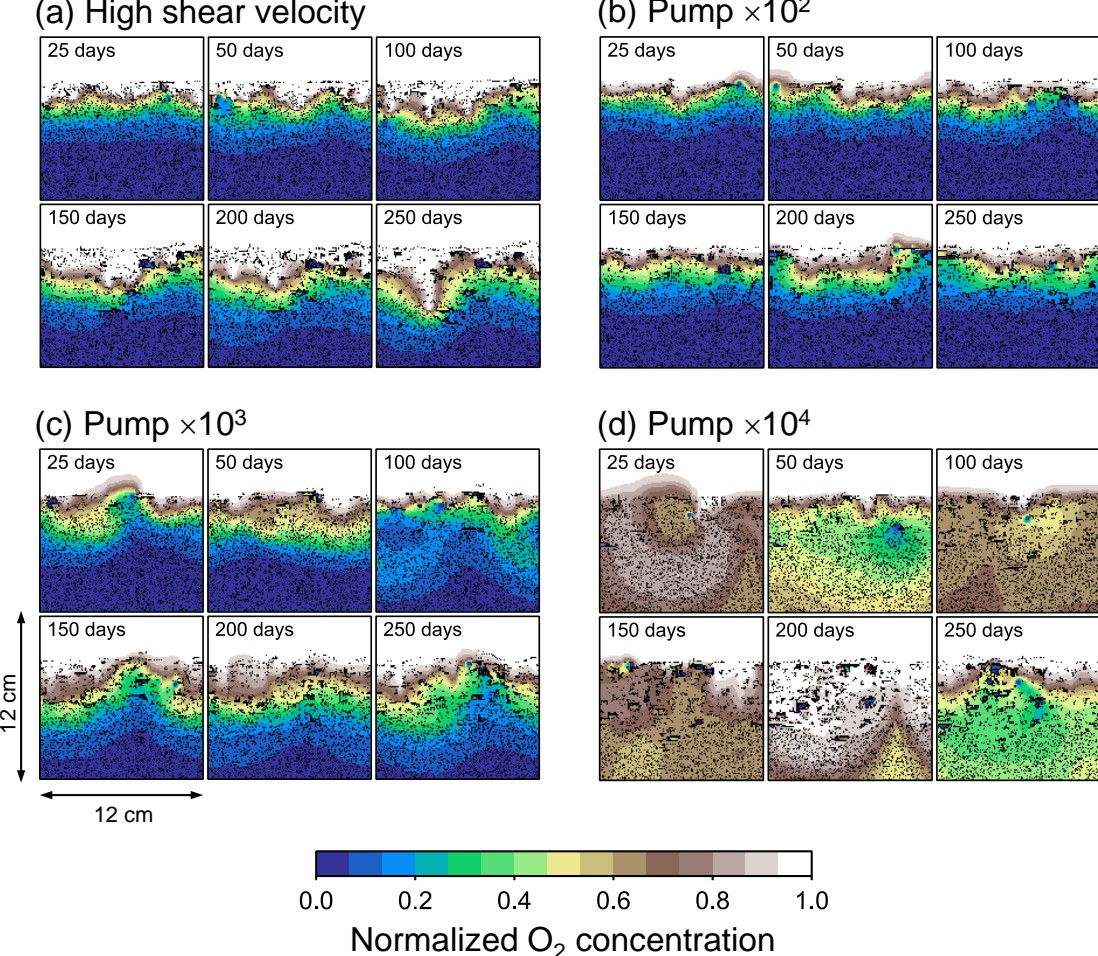

(a) High shear velocity

| 25 days | 50 days | 100 days |
| 150 days | 200 days | 250 days |

(b) Pump $\times 10^2$

| 25 days | 50 days | 100 days |
| 150 days | 200 days | 250 days |

(c) Pump $\times 10^3$

| 25 days | 50 days | 100 days |
| 150 days | 200 days | 250 days |

12 cm
12 cm

(d) Pump $\times 10^4$

| 25 days | 50 days | 100 days |
| 150 days | 200 days | 250 days |

0.0   0.2   0.4   0.6   0.8   1.0

Normalized $O_2$ concentration

Figure E3

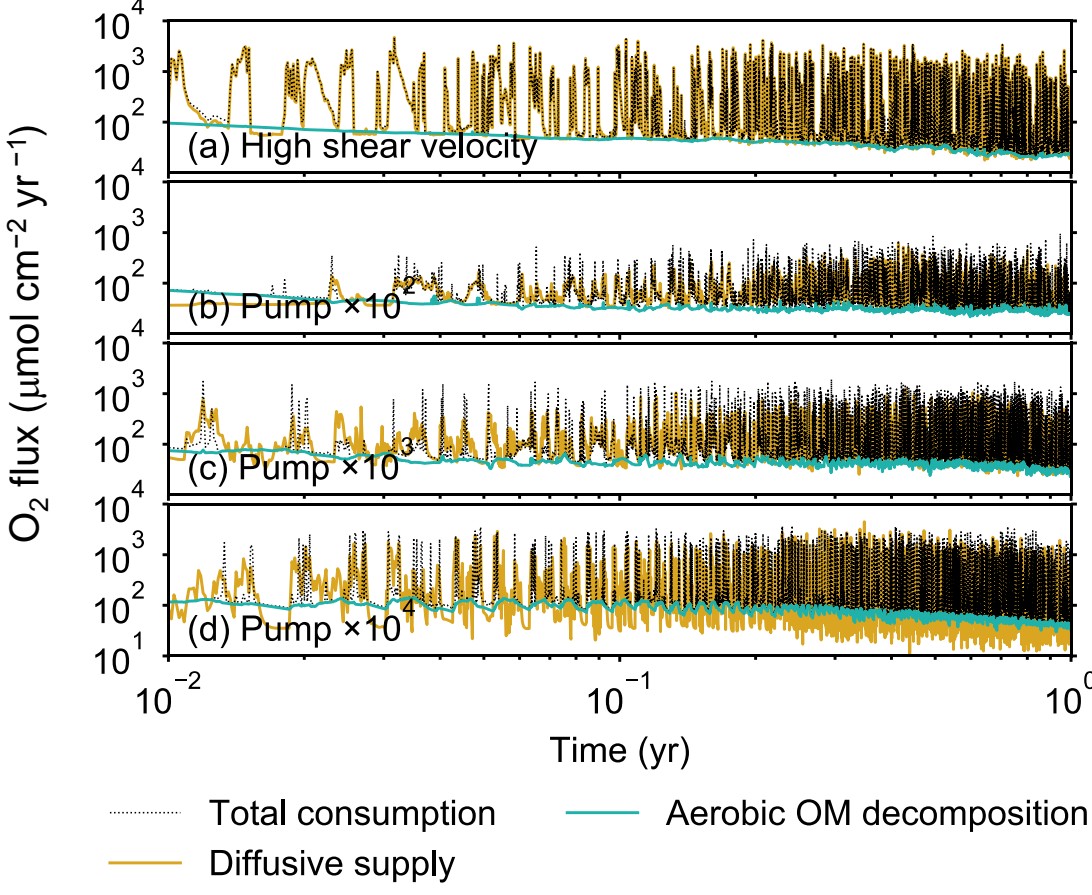

Figure E4

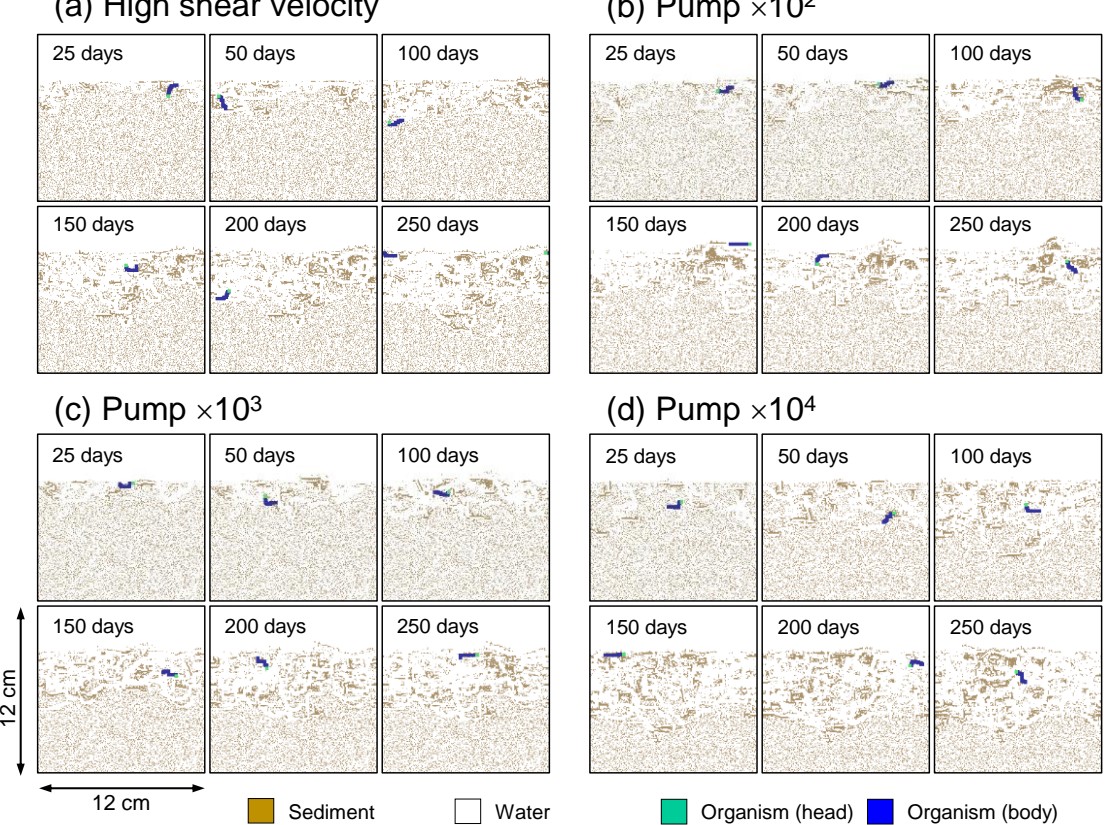

(a) High shear velocity

(b) Pump ×10²

(c) Pump ×10³

(d) Pump ×10⁴

25 days  50 days  100 days
150 days  200 days  250 days

12 cm

12 cm

Sediment  Water  Organism (head)  Organism (body)

Figure E5

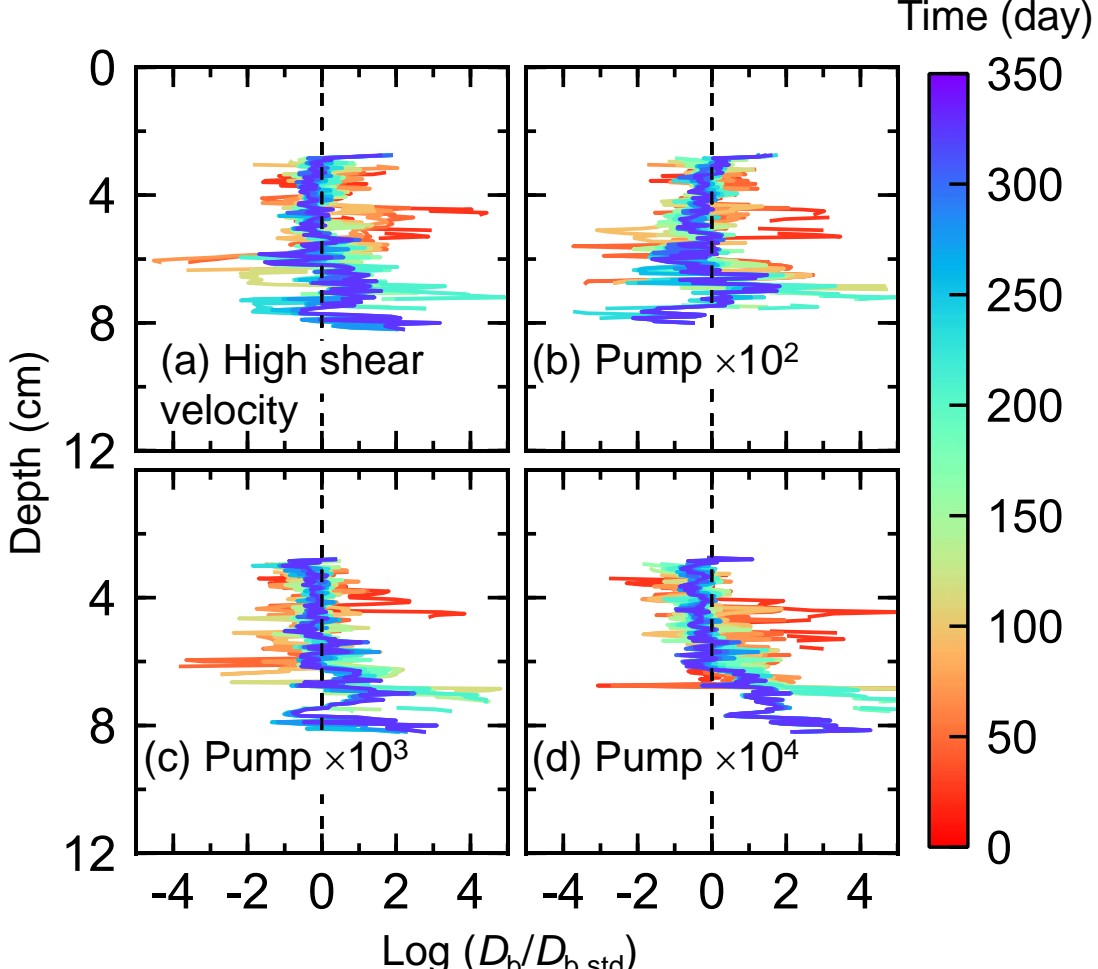