# Peer review of "A lattice-automaton bioturbation simulator with coupled physics, chemistry, and biology in marine sediments (eLABS v0.2)"

_Geoscientific Model Development, 2019_

## Referee Comment (RC1) · Anonymous Referee #1 · 17 May 2019

Kanzaki and co-authors present a bioturbation simulator that is based on the LABS model, originally presented in Boudreau et al. 2001 and Choi et al. 2002. They expand that model to include the simulation of water flow, O2 and organic matter distributions. The main motivation is to establish a framework that has a mechanistic foundation to interpret the geologic record of trace fossils and provide an explanation for empirical relationships observed in the modern ocean between bioturbation and sediment properties. The manuscript is well written and organized, and the consistent presentation of the results across the 3 sets of figures make it easy for the reader to follow. They authors provide interesting results and shows some feedbacks (e.g. page 13, line 10) that illustrate the usefulness of the modeling approach, which I consider suitable for

[Figure]

GMD. There are, however, also a few shortcomings.

Validation: The model presentation introduces features that generally make sense, such as behavior tied to the presence of O2 or the reactivity of organic matter. However, what is lacking is a rigorous validation of the results, which of course is rather challenging. A key concern is the limited discussion what organisms the model represents. LABS was a novel way of looking at organisms that move sediment around. Are these the same organisms that govern the distribution of O2? Arguably, the most pronounced impacts on O2 in contemporary sediments are not caused by moving around sediment blocks but by the flushing of burrows, injection of fluid into the subsurface or similar activities. Thus, organisms other than the ones studied with eLABS, or other activities (e.g. pumping, rather than directional movement of infauna) by the organism studied may set the O2 distribution. Under these conditions, feedbacks other than that between movement and oxygenation considered in eLABS (which in the applications shown here possibly represents the 'simple small deposit feeders, resembling capitellids' referred to in Choi et al. 2002) may be dominant, and limit the ability of the model to capture the connections between food, air and organism.

Implementation: The calculation of the oxygen concentration is conducted on a grid that is occupied by water/organism particles with a general advection-diffusion-reaction equation (e.g., Boudreau, 1997) (EQ. 2). This is necessary as in the model there is no O2 in the (zero porosity) in the solid cells, yet no organic matter (M) in the fluid phase. However, nothing is stated what the size of this large scale grid is that represents average solid and fluid concentrations, or how exactly u and D are computed in the sediment.

Results: How robust are the findings, shown for example in Figure 7? What is the uncertainty in the stochastic default simulation? This is particularly important as results are shown in several panels relative to the default. For example, Fig11b shows an increase in Db at depth relative to the default, at a depth where Db is small. Is this consistently found for multiple realizations of the default simulation? If not, consider

establishing uncertainty estimates. And is any of the patterns seen an effect of the boundaries (i.e. too small domain size)? On page 12, at the end of the first paragraph, it is stated that "When we remove the advective flow from the calculations, i.e., simulation (f), the resultant oxygen profiles, fluxes, burrow geometry and biodiffusion coefficient are generally similar to those with the default settings (panels (a) and (f) of Figs. 4, 5, 6, 7). Accordingly, with the assumptions adopted in the present study, advective water flow has only insignificant influences on bioturbation." If one follows the bioturbation definition of Kristensen et al. 2012, MEPS ("all transport processes carried out by animals that directly or indirectly affect sediment matrices.") I'd agree that at the low velocities resulting from with the movement of macrofauna, fluid flow is not important for solid phase distribution. However, there is ample observational evidence (documented in several of the papers cited) that shows the importance of bioadvective flow for solute distributions (O2 profiles) and fluxes.

Summary: The summary/conclusions points to what is missing in the paper, namely a a rigorous testing against data (page 15, line 16). Overall, I consider this an interesting paper and valuable expansion on an existing model. It clearly is a framework to test scientific hypotheses. The potential is undoubtedly there, but the feasibility of establishing the mechanistic explanations for empirical relationships largely remains to be demonstrated and will be a significant undertaking.

additional notes:

page 5/ line 20: avoiding oxygen depleted conditions does not necessarily translate into moving against the O2 gradient. This seems to align with the statement on 14/34 that stresses the importance of food availability

9/24: it says that the infaunal respiration flux = total O2 consumption flux - O2 flux by aerobic decomposition. I may be thrown off by the terminology (flux, rates), but what about non-steady state effects, and advective or diffusive transport?

10/18: it states that organic matter concentration is assumed to decrease with depth.

However, on page 3, deterministic calculations of organic matter distributions are discussed. Please clarify

11/11: how are permeability and porosity connected in the model? If this is explained in LABS papers cite the relevant publication.

Consider discussing the role of predation on burrowing, in addition to the role of organic matter and oxygen

Is figure 3 necessary?
* * *

---

## Referee Comment (RC2) · Anonymous Referee #2 · 5 Jun 2019

This paper presents an extension to the LABS model. In the new model, the authors implemented the calculation of water flow and oxygen and organic matter concentration filed. It is useful for theoretical investigations into the interplay between biological, physical and chemical factors influencing sediment bioturbation. The manuscript is well written and organized. This is an important work that merits publication, and GMD is an appropriate journal for this work. However, the manuscript needs substantial improvement of the presentation before it can be recommended for publication. The two main problems are (1) clarity in the presentation of the model and (2) generality of the results.

[Figure]

1) The distinct advantage of eLABS is the addition of water flow, which could couple the overlying water with sediment continuously. However, it is too rough for the model description and therefore its validity cannot be judged. For example, the proper treatment of the moving boundary when the sediment was moved by organisms is essential to get the flow field. Only constant flows imposed in the head or tail of the organisms may not sufficient. The pressure and velocity along the moving sediment is also important. The accuracy of boundary condition also largely depends on the grid resolution, especially in this case that the sediment occupied only one grid. Therefore, grid refinement is necessary to check the error bar. Similarly, the treatment of the boundary condition of oxygen on the moving sediment has to be careful to ensure the mass conservation. One could check the mass conservation of oxygen by turn off the oxygen consumption rate. The implicit and explicit finite-difference method as well as the boundary condition could be presented in the appendix to guide readers easier. Before considering the effects of biological, physical and chemical factors on bioturbation, the verification of numerical implement could also be put in the appendix.

Organic matter is generally located in the solid particles and oxygen in water. How to calculate the rate with Eqs.(4) and (5) when OM and O2 are not at the same cell?

2) The model was run only once for one case study, which loose its generality. Lattice-automaton contains stochastic processes. It is better to consider different initial distribution the sediment and random generator for animal move. The ensemble averaged effects of biological, chemical and physical parameters on oxygen fluxes and rates of mixing in ocean sediments could provide a mechanistic explanation for empirical relationships observed in the modern ocean sediments, which is much more useful. Otherwise, it becomes meaningless due to large uncertainty and randomness. According to the model setup, the running will lead to a steady state. It the time is within geology years, one could compare the results in the steady state, which is comparable with the observed empirical relation.

–other notes–

Page 5 /line 28: Non-local mixing of water (bio-irrigation) by infauna is already represented in LABS. In the original LABS, there is only non-local mixing of sediment and no bio-irrigation is presented.

6/1-2: Many readers may not familiar with "marker and cell method". It is better to give some details. The references cited here (Hoffmann and Chiang 2000; Manwart et al 2002; Meysmann et al. 2005,2006b,2007; Volkenborn et al. 2012) are not properly.

6/11: For fluid the name "no-vertical-flux boundary conditions" is not used. Instead, slip boundary condition is common used. "Non-slip boundary" should be replaced by "No-slip boundary". "left and right boundaries are continuous" could be simply replaced by "periodic boundary condition".

7/7: The shear velocity is usually resulted from a turbulence flow in the overlying water. Within the lowest portion of the planetary boundary layer a semi-empirical log velocity profile is used. However, in this paper, there is any external flow in the water and shear velocity lose its meaning.

9/32: Does "biodiffusion coefficients in the present study are obtained by calculating average values..." mean that Db is depth independent. Actually, Db depends on the sediment depth.

11/5: The unexpectedly larger of biodiffusion coefficient at ∼7 to 8 cm depths results from only one sample run. If one runs more samples and average them, I think the "unexpectedly" will be disappear. It is not from non-local mixing.

12/3: The authors mentioned that "advective water flow has only insignificant influences on bioturbation". In fact, people are more interested in the effect of bioturbation on the advective water flow and thereafter the bioirrigation, which might significantly change oxygen flux.

---

## Author Comment (AC1) · 20 Aug 2019

**Response to Anonymous Referee #1**

We express our gratitude to Anonymous Referee #1 for his/her useful comments. Our response to the reviewer's comments and the corresponding revision are described in detail below. The numbers of pages, lines, equations, tables and figures are those in the revised manuscript unless otherwise described.

Comment 1:
"Validation: The model presentation introduces features that generally make sense, such as behavior tied to the presence of O2 or the reactivity of organic matter. However, what is lacking is a rigorous validation of the results, which of course is rather challenging. A key concern is the limited discussion what organisms the model represents. LABS was a novel way of looking at organisms that move sediment around. Are these the same organisms that govern the distribution of O2? Arguably, the most pronounced impacts on O2 in contemporary sediments are not caused by moving around sediment blocks but by the flushing of burrows, injection of fluid into the subsurface or similar activities. Thus, organisms other than the ones studied with eLABS, or other activities (e.g. pumping, rather than directional movement of infauna) by the organism studied may set the O2 distribution. Under these conditions, feedbacks other than that between movement and oxygenation considered in eLABS (which in the applications shown here possibly represents the 'simple small deposit feeders, resembling capitellids' referred to in Choi et al. 2002) may be dominant, and limit the ability of the model to capture the connections between food, air and organism."

Response:
Simulated values of biodiffusion coefficients and oxygen fluxes are well within the observed ranges (Section 3.1). Thus, we consider that eLABS yields reasonable results, comparable to modern observations. While we agree that further confirmation of the model's validity would be desirable, experimental data that can be directly compared in detail with the model's settings and results is not available and thus further confirmation of the model's validity is not currently feasible.

   According to Jumars and Wheatcroft (1989, In: Productivity of the Ocean: Present and Past (eds. W. H. Berger, V. S. Smetacek and G. Wefer), pp. 235-253), deposit feeders are responsible for most of sediment bioturbation for most of the time. Therefore, we consider the choice of the organism for the present study reasonable (a deposit feeder which resembles a capitellid and whose body size and ingestion rate are within the observed ranges described by Cammen (1980), Kemp (1987) and Lopez and Levinton (1987)). Although the details of the organisms (species) that are responsible for bioturbation can change with time and space (e.g., Aller, 2001; Kristensen et al., 2012; Tarhan et al., 2015), a strength of eLABS is that we can also change the burrow density and feeding rates by changing rules that govern organism behavior and biological parameters of the

organism (e.g., Fig. 2).

Flushing of burrows and injection of fluid into the subsurface are likely to be important in permeable sediment (e.g., Huettel and Webster, 2001). Such phenomena require relatively strong water flows above sediment. However, direct simulation of such strong flows would be difficult because a higher grid resolution would be required. Therefore, instead of direct simulation, we formulated the governing equation for oxygen with an effective diffusion term so that it includes the effect of oxygen mixing caused by turbulent flows above the seawater-sediment interface, i.e., an eddy diffusion term in addition to the term for molecular diffusion. The strength of eddy diffusion can be changed with shear velocity (e.g., Fig. 4e), which is related to the current strength in the water column above the modeled sediment (e.g., Pope et al., 2006).

To simply and implicitly enable a pumping action, the model has been updated (v0.2) so that the constant water flows imposed when an organism ingests or egests sediment particles can be increased by an arbitrary factor. Simulations where the water flows imposed at the time of ingestion/egestion are increased by factors between $10^2$ and $10^4$ suggest that greater oxygen can be introduced into sediment as a result of enhanced advective water flows and that this causes a significant increase in infaunal respiration flux, similar to the case with a high shear velocity. This similarity is reasonable given that both simulations implicitly employ increased contributions of advective water flows to oxygen mixing into sediment.

Changes in manuscript (Page numbers/Line numbers):
The need for a further validation of the model was additionally described in Section 4 (P/16/L32-P17/L1-2).

We mention that a variety of organisms can be responsible for bioturbation and, because of this, LABS is beneficial due to its ability to simulate various biological parameters that control organism behavior (P2/L25-27, P4/L31).

We have now clarified that flushing of burrows and injection of fluid into the subsurface can be implicitly simulated by increasing shear velocity (P12/L19-21).

We have added an appendix which shows results of simulations where water flows imposed at the time of ingestion/egestion are increased by factors between $10^2$ and $10^4$ (Appendix E, P19/L2-14). We mention that the simulation where advective flows contribute significantly to oxygen mixing leads to bioturbation results similar to those in the case where shear velocity is high and reference Appendix E (P12/L30-31).

Comment 2:
"Implementation: The calculation of the oxygen concentration is conducted on a grid that is occupied by water/organism particles with a general advection-diffusion-reaction equation (e.g., Boudreau, 1997) (EQ. 2). This is necessary as in the model there is no O2 in the (zero porosity) in the solid cells,

yet no organic matter (M) in the fluid phase. However, nothing is stated what the size of this large scale grid is that represents average solid and fluid concentrations, or how exactly u and D are computed in the sediment."

Response:

The calculation of oxygen and organic matter is conducted directly on the eLABS grid, i.e., by using individual eLABS grid cells as the finite difference grid cells. Thus, a value of organic matter concentration is assigned to each sediment grid cell while an oxygen concentration is assigned to each water/organism grid cell. Calculation of organic matter degradation is conducted wherever two grid cells of water/organism and sediment are located next to one another and degradation rate information is shared between the two grid cells to be reflected in the calculations of organic matter and oxygen concentrations.

      Similarly, a set of flow velocities and an effective diffusion coefficient is assigned to each water/organism grid cell. Please note that because the water flow is simulated by the marker and cell method, four velocities are assigned on the edges of each grid cell while one pressure is defined at the center of each grid cell (e.g., Hoffmann and Chiang, 2000).

Changes in manuscript (Page numbers/Line numbers):

We have added explanations of the calculations of water flow, organic matter and oxygen (as in our response above) to the revised manuscript (P5/L33-P6/L3, P7/L31-P8/L5, P8/L24-P9/L2).

Comment 3:

"Results: How robust are the findings, shown for example in Figure 7? What is the uncertainty in the stochastic default simulation? This is particularly important as results are shown in several panels relative to the default. For example, Fig11b shows an increase in Db at depth relative to the default, at a depth where Db is small. Is this consistently found for multiple realizations of the default simulation? If not, consider establishing uncertainty estimates. And is any of the patterns seen an effect of the boundaries (i.e. too small domain size)? On page 12, at the end of the first paragraph, it is stated that "When we remove the advective flow from the calculations, i.e., simulation (f), the resultant oxygen profiles, fluxes, burrow geometry and biodiffusion coefficient are generally similar to those with the default settings (panels (a) and (f) of Figs. 4, 5, 6, 7). Accordingly, with the assumptions adopted in the present study, advective water flow has only insignificant influences on bioturbation." If one follows the bioturbation definition of Kristensen et al. 2012, MEPS ("all transport processes carried out by animals that directly or indirectly affect sediment matrices.") I'd agree that at the low velocities resulting from with the movement of macrofauna, fluid flow is not important for solid phase distribution. However, there is ample observational evidence (documented in several of the papers cited) that shows the importance of bioadvective flow for solute distributions

(O2 profiles) and fluxes."

Response:

We ran 5 runs for each simulation to evaluate the contribution of stochastic processes to the bioturbation results in Section 3.

Increasing the size of calculation domain does not significantly change the results. Please note that the left and right boundaries adopt the periodic boundary condition (i.e., they are continuous) and also, if we increase the modeled sediment grid width (12 cm in the default setting) by a factor of 2, then the number of organisms must be increased by the same factor to maintain the same population density of organisms in the sediment. We conducted a simulation where the calculation domain is wider by a factor of 2 and populated by two organisms of the same properties and with otherwise default settings. The results using the wider calculation domain and correspondingly increased number of deposit feeders are not significantly different from those using the default settings.

We agree that with a pumping action (e.g., Meysman et al., 2005), advective flow caused by infauna can become more important. As in our response to Comment #1 by Anonymous Referee #1, we ran simulations with increased biologically induced water flow to implicitly account for pumping action in eLABS. According to these simulations, results from increasing the contribution of advection to oxygen mixing within sediment are similar to those using a high shear velocity. Nonetheless, further development of eLABS is necessary to explicitly simulate pumping action. Please also see our response to Comment 1 by Anonymous Referee #1.

Changes in manuscript (Page numbers/Line numbers):

We have added a description of stochastic effects in Section 3 (P9/L31-P10/L3, P13/L21-22, P14/L12-13). Where necessary, we also modified descriptions in Section 3 indicating the multiple results for each simulation (P12/L1-2, P12/L13-14, P12/L27-28, P13/L28, P14/L12, P15/L23-24).

We have added an appendix which shows results from 5 runs of the default simulation to illustrate the effects of stochastic animal behavior in eLABS (Appendix D, P18/L25-29) and another appendix which shows the results from a simulation where a wider calculation domain is assumed (Appendix B, P18/L1-6).

We have added still another appendix to show results from simulations where biologically induced water flows are increased (Appendix E, P19/L2-14). We also mention that simulations with a high contribution of advection to oxygen mixing yield similar bioturbation results to those from a simulation with a high shear velocity and refer to Appendix E (P12/L30-31). Please also see our changes in manuscript in response to Comment 1 by Anonymous Referee #1.

Comment 4:

"Summary: The summary/conclusions points to what is missing in the paper, namely a rigorous testing against data (page 15, line 16). Overall, I consider this an interesting paper and valuable expansion on an existing model. It clearly is a framework to test scientific hypotheses. The potential is undoubtedly there, but the feasibility of establishing the mechanistic explanations for empirical relationships largely remains to be demonstrated and will be a significant undertaking."

Response:
We have addressed the issue of a more rigorous testing against data in our response to Comment 1 by Anonymous Referee #1.

Changes in manuscript:
Please see our changes in manuscript in response to Comment 1 by Anonymous Referee #1.

Additional note 1:
"page 5/ line 20: avoiding oxygen depleted conditions does not necessarily translate into moving against the O2 gradient. This seems to align with the statement on 14/34 that stresses the importance of food availability"

Response:
Agreed.

Changes in manuscript (Page numbers/Line numbers):
We have added 'Note, however, that the stochastic behavior may occasionally lead organisms to unfavorable locations with respect to food and/or oxygen availability.' (P5/L23-24).

Additional note 2:
"9/24: it says that the infaunal respiration flux = total O2 consumption flux - O2 flux by aerobic decomposition. I may be thrown off by the terminology (flux, rates), but what about non-steady state effects, and advective or diffusive transport?"

Response:
We use the term 'flux' to refer to a mass change in unit area per unit time (e.g., mol $cm^{-2}$ $yr^{-1}$), while the term 'rate' was used to refer to a mass change in unit volume per unit time (e.g., mol $cm^{-3}$ $yr^{-1}$). Thus, depth integration of consumption rates within sediment provides the oxygen consumption flux.

In eLABS, oxygen fluxes caused by changes in the total amount of oxygen within the calculation domain, aerobic organic matter degradation, infaunal respiration, advection and effective

diffusion (i.e., molecular plus eddy diffusion) are calculated at each time step, as well as the residual oxygen flux as sum of the above fluxes to evaluate the calculation error (the residual flux must be close to zero if the mass of oxygen is conserved). The first flux, i.e., oxygen flux caused by changes in the total amount of oxygen within the calculation domain, represents the flux that is significant when the porewater chemistry is far from steady state. This non-steady-state flux can be recognized as a deviation of total oxygen supply (dominated by diffusive supply; orange curves in Figs. 5, 9 and 13) from the total oxygen consumption (black dotted curves), which is significant over dozens of model days from the start of simulations. After this initial period, the non-steady-state flux becomes relatively insignificant.

Changes in manuscript (Page numbers/Line numbers):
We have added a description similar to that above in the revised manuscript (P10/L18-19, P17/L11-26).

Additional note 3:
"10/18: it states that organic matter concentration is assumed to decrease with depth. However, on page 3, deterministic calculations of organic matter distributions are discussed. Please clarify"

Response:
A decrease of organic matter concentration with depth is adopted as the initial condition for organic matter. Changes in concentration of organic matter from this initial condition are deterministically calculated as explained in Section 2.3 of the previous manuscript.

Changes in manuscript (Page numbers/Line numbers):
We have modified descriptions in Section 2.3 to clarify the model calculation and assumption (P8/L24-26).

Additional note 4:
"11/11: how are permeability and porosity connected in the model? If this is explained in LABS papers cite the relevant publication."

Response:
Porosity is assumed in eLABS, based on which sediment particles are randomly distributed. Permeability can be calculated by using the water flow calculation, solving the Navier-Stokes equation using sediment pore geometry that is determined by distributions of sediment particles (e.g., Manwart et al., 2002). Because the water flow calculation is one of the new features of eLABS, i.e.,

not included in LABS, there are no LABS papers which describe the relationships between permeability and porosity.

Changes in manuscript (Page numbers/Line numbers):
We added an appendix which explains that permeability can be calculated by solving the Navier-Stokes equation once the pore geometry (i.e., eLABS grid) is given (Appendix C, P18/L8-23).

Additional note 5:
"Consider discussing the role of predation on burrowing, in addition to the role of organic matter and oxygen"

Response:
The role of predation on burrowing (e.g., Posey, 1986, J. Exp. Mar. Bio. Ecol. 103, 143-161) cannot be explicitly discussed because LABS, on which eLABS was developed, does not include predation as an animal behavior. In ongoing developments to LABS, however, we have allowed changes in animal population as well as sizes of individual benthos based only on food supply [Kanzaki et al., in prep.]. Hence, we will be able to realize predation in future versions of LABS, but this is beyond the scope of our present study.

Changes in manuscript (Page numbers/Line numbers):
We have mentioned predation as a feature to be realized in eLABS in future developments in Section 4 (P16/L30).

Additional note 6:
"Is figure 3 necessary?"

Response:
We consider Fig. 3 necessary because the calculation of a water flow field based on animal behavior is a new feature of eLABS first presented in this study. Although advective water flow was found not to be significant when using the default setting for the present study, advective flow can become more important in simulations where biologically induced water flows are increased.

Changes in manuscript (Page numbers/Line numbers):
We have added an appendix which shows simulations where biologically induced water flows are increased (Appendix E, P19/L2-14). We mention that the conditions where the advective water flow

is important for oxygen mixing leads to similar bioturbation results compared to those in a simulation with high shear velocity (P12/L30-31).

---

## Author Comment (AC2) · 20 Aug 2019

**Response to Anonymous Referee #2**

We express our gratitude to Anonymous Referee #2 for his/her useful comments. Our response to the reviewer's comments and the corresponding revisions to our manuscript are described in detail below. The numbers of pages, lines, equations, tables and figures are those in the revised manuscript unless otherwise noted.

Comment 1:
"The distinct advantage of eLABS is the addition of water flow, which could couple the overlying water with sediment continuously. However, it is too rough for the model description and therefore its validity cannot be judged. For example, the proper treatment of the moving boundary when the sediment was moved by organisms is essential to get the flow field. Only constant flows imposed in the head or tail of the organisms may not sufficient. The pressure and velocity along the moving sediment is also important. The accuracy of boundary condition also largely depends on the grid resolution, especially in this case that the sediment occupied only one grid. Therefore, grid refinement is necessary to check the error bar. Similarly, the treatment of the boundary condition of oxygen on the moving sediment has to be careful to ensure the mass conservation. One could check the mass conservation of oxygen by turn off the oxygen consumption rate. The implicit and explicit finite-difference method as well as the boundary condition could be presented in the appendix to guide readers easier. Before considering the effects of biological, physical and chemical factors on bioturbation, the verification of numerical implement could also be put in the appendix. Organic matter is generally located in the solid particles and oxygen in water. How to calculate the rate with Eqs.(4) and (5) when OM and O2 are not at the same cell?"

Response:
The imposed constant flow rate is calculated based on the momentum changes of sediment particles moved directly by organisms as well as those through organisms' movements. Thus, temporal changes in boundary conditions caused by sediment particle displacements through organisms' movements, as well as those caused by organisms' movements themselves are adequately accounted for in the present study. We also updated eLABS (now v0.2) to enable the water flow simulation on a higher grid resolution than that on which animal behavior, oxygen and organic matter are simulated (e.g., 480×480 grid for water flow simulation vs. 240×240 grid for calculations of animal behavior, oxygen and organic matter). By this, sediment particles are occupied with more than one grid cell during the flow simulation. The default simulation for 1 model year with an increased grid resolution (480×480) for the water flow calculation shows no significant difference from that with the default resolution. Accordingly, we consider that the default resolution of the eLABS grid is sufficient for producing reasonable bioturbation results.

eLABS calculates and reports mass fluxes of oxygen caused by temporal change in the total amount of oxygen within the calculation domain, aerobic organic matter degradation, infaunal respiration, molecular plus eddy diffusion and advection, as well as the residual flux as a sum of the above fluxes, which ideally should be zero, at every time step. Numerical solutions by the finite difference method generate a non-zero residual flux of oxygen but this flux is generally insignificant. In eLABS v0.2, the absolute value of residual flux is generally less than $\sim 10^{-5}$ % of the absolute total oxygen consumption.

When locations of sediment and water particles are exchanged through pushing by the organism, their oxygen concentrations are also exchanged. The boundary condition for the oxygen calculation is updated based on pore geometry at every time step. Moving boundary conditions are thus accounted for. We also confirmed that moving boundaries do not cause any changes in oxygen concentrations by observing no changes in oxygen concentrations in a simulation where oxygen consumption and ingestion/egestion by infauna are excluded. Please note that when including ingestion, even assuming zero oxygen consumption, there can still be oxygen fluxes because sediment particles removed during ingestion are replaced by water particles with zero oxygen concentration. Although this does not cause any change in the oxygen amount at the time of replacement, more oxygen can be transported toward the replaced water particles in the following time steps, causing an oxygen transport flux. This oxygen flux accompanying ingestion is inevitable and insignificant ($\sim$1%) compared to oxygen consumption by respiration and organic matter degradation.

Organic matter decomposition in a given sediment grid cell in a given time step is calculated by first examining adjacent grid cells. We regard two grid cells as connected only when they share one cell edge (i.e., one grid cell has four connected neighbor cells at maximum). When one sediment grid cell is connected to a water/organism grid cell, then the degradation rate of organic matter is calculated using the oxygen concentration of the water/organism grid cell and the organic matter concentration of the sediment grid cell. This degradation rate information is shared with the water/organism grid cell to be used for the oxygen concentration calculation. When there are multiple water/organism grid cells connected to a sediment grid cell, degradation rates of organic matter are similarly calculated. The connected water/organism grid cells thus obtain information on organic matter degradation caused by reactions between organic matter in the connected sediment grid cell and oxygen in their own grid cells, while the sediment grid cell has information on degradation caused by the sum of reactions with all the connected water/organism grid cells. The above calculation is repeated for all sediment grid cells, by which each water/organism grid cell obtains information on organic matter degradation from all connected sediment grid cells.

Changes in manuscript (Page numbers/Line numbers):
We have added further explanation for the calculations of water flow, organic matter and oxygen based on the description presented in our response to the comment just above (P5/L33-P6/L1-3,

P6/L13, P7/L31-P8/L5, P8/L24-P9/L2). We also added an appendix that shows several simulations including those without oxygen consumption to verify the conservation of oxygen in eLABS (Appendix A, P17/L11-26).

We did not add an appendix to present the finite difference method because the finite difference method is explained in many textbooks (e.g., Hoffmann and Chiang, 2000). Nonetheless, we added more explanations on the finite difference method in Section 2 (P7/L31-32). Although the reviewer suggested that boundary conditions may be described in appendices, we have already presented boundary conditions in Section 2. Accordingly, instead of adding an appendix, we added additional explanations of boundary conditions where necessary based on the comment by the reviewer (please also see our response to Other note 3 by Anonymous Referee #2) (P6/L10, P6/L11, P7/L26).

Comment 2:
"The model was run only once for one case study, which loose its generality. Lattice-automaton contains stochastic processes. It is better to consider different initial distribution the sediment and random generator for animal move. The ensemble averaged effects of biological, chemical and physical parameters on oxygen fluxes and rates of mixing in ocean sediments could provide a mechanistic explanation for empirical relationships observed in the modern ocean sediments, which is much more useful. Otherwise, it becomes meaningless due to large uncertainty and randomness. According to the model setup, the running will lead to a steady state. It the time is within geology years, one could compare the results in the steady state, which is comparable with the observed empirical relation."

Response:
We agree with the reviewer that ensembles of simulations will be useful to evaluate how much the stochastic process contributes to the bioturbation results. We ran each simulation five times to evaluate the contribution of stochastic processes. Different initial distributions of sediment can be considered by these multiple runs because initial distributions are randomly determined even assuming the same porosity. Next, we conducted simulations with different sediment porosity, which also creates different sediment particles distributions.

Although it is desirable to run the model on geological time scales (e.g., > 1000 years), this is difficult because of the relatively heavy calculation (a 1-yr simulation with the default setting takes ~2 days, meaning that a 1000-yr simulation will take ~2000 days or ~5 years). Nonetheless, porewater chemistry reaches steady state relatively fast compared to solid phase species (Archer et al., 2002). We also assume a decrease of organic matter concentration with depth as the initial condition, which may be regarded as steady-state distribution of organic matter (e.g., Van Cappellen and Wang, 1996). Then, the simulated profiles may be regarded to be close to those in steady state and be compared with modern sediment observations.

Changes in manuscript (Page numbers/Line numbers):

We added an appendix which shows additional simulations to illustrate the contribution of the stochastic process to bioturbation results (Appendix D, P18/L25-29). Where necessary, descriptions in Section 3 were also modified (P12/L1-2, P12/L13-14, P12/L27-28, P13/L28, P14/L12, P15/L23-24).

We added an explanation that assuming a decrease of organic matter with depth may help the model reach a steady state despite a relatively short run-time (P8/L24-25).

Other note 1:

"Page 5 /line 28: Non-local mixing of water (bio-irrigation) by infauna is already represented in LABS. In the original LABS, there is only non-local mixing of sediment and no bio-irrigation is presented."

Response:

In LABS, when a line of sediment particles is pushed by an organism, a water particle that exists at the far end of the line of pushed sediment particles is moved to the location where the sediment particle at the nearest end of the line of the pushed sediment particles was located before pushing (Choi et al., 2002). Thus, non-local mixing of water particles is implicitly realized in LABS. However, individual water particles are not tracked and since there are no tracers on water particles, the effect of non-local mixing of water cannot be evaluated in LABS.

Changes in manuscript (Page numbers/Line numbers):

We deleted '(bio-irrigation)' in the relevant sentence because non-local mixing of water in LABS is caused through particle displacements (please see above), but not directly by biological processes such as water-pumping action (Meysman et al., 2005) (P5/L29).

Other note 2:

"6/1-2: Many readers may not familiar with "marker and cell method". It is better to give some details. The references cited here (Hoffmann and Chiang 2000; Manwart et al 2002; Meysmann et al. 2005,2006b,2007; Volkenborn et al. 2012) are not properly."

Response:

We agree with the reviewer that the marker and cell method may not be familiar with all readers. Hoffmann and Chiang (2000) provided an explanation of the marker and cell method in their Chapter 8.7.1, and Manwart et al. (2002) utilized a marker and cell grid for their finite difference solution of

the Navier-Stokes equation. Thus, we consider these two references appropriate. Also, Meysman et al. (2005, 2006b, 2007) and Volkenborn et al. (2012) conducted water flow simulations within sediments based on Darcy and Navier-Stokes/Brinkman equations, respectively. Because these equations are based on the same fluid mechanics (e.g., Das, 1997, J. Can. Petrol. Technol., 36, 57-59), we also consider these references relevant.

Changes in manuscript (Page numbers/Line numbers):
We added an explanation for the marker and cell method and deleted Meysman et al. (2005, 2006b, 2007) and Volkenborn et al. (2012) from the relevant sentence (P5/L33-P6/L3).

Other note 3:
'6/11: For fluid the name "no-vertical-flux boundary conditions" is not used. Instead, slip boundary condition is common used. "Non-slip boundary" should be replaced by "No-slip boundary". "left and right boundaries are continuous" could be simply replaced by "periodic boundary condition".'

Response:
We meant zero pressure gradients by "no-vertical-flux boundary conditions". We agree with the reviewer on the other terminology.

Changes in manuscript (Page numbers/Line numbers):
We added '(zero pressure gradients)' in the relevant sentence (P6/L10). We used 'no-slip boundary' instead of 'non-slip boundary', and 'periodic boundary condition' to describe the left and right boundary conditions in the revised manuscript (P6/L11).

Other note 4:
"7/7: The shear velocity is usually resulted from a turbulence flow in the overlying water. Within the lowest portion of the planetary boundary layer a semi-empirical log velocity profile is used. However, in this paper, there is any external flow in the water and shear velocity lose its meaning."

Response:
The reason we introduced the shear velocity is to implement oxygen mixing caused by eddy diffusion to represent oxygen transport through turbulent flow above sediment (e.g., Volkenborn et al., 2002). Direct simulations of turbulent flow would require a gird with a higher resolution and increase the calculation time. By introducing the eddy diffusion term, which is formulated with the shear velocity (a higher shear velocity leads to a stronger eddy diffusion), the intensity of the current above sediments can be implicitly considered because an increase in the current intensity should be

accompanied with an increase in shear velocity (e.g., Pope et al., 2006).

Changes in manuscript (Page numbers/Line numbers):
We have added further explanations of the shear velocity in the revised manuscript (P12/L19-21).

Other note 5:
'9/32: Does "biodiffusion coefficients in the present study are obtained by calculating average values..." mean that Db is depth independent. Actually, Db depends on the sediment depth.'

Response:
Biodiffusion coefficients ($D_b$) can change with the sediment depth (Figs. 7, 11 and 15). We did not make any assumptions regarding the depth-$D_b$ relationship. The $D_b$ values are calculated based on displacements of sediment particles. Because averages of sediment particles displacements in individual depth layers are used for the calculation of $D_b$ with depth, the calculated $D_b$ values can change with sediment depth.

Changes in manuscript (Page numbers/Line numbers):
We have added further explanation of the calculation of $D_b$ (P10/L28-29).

Other note 6:
'11/5: The unexpectedly larger of biodiffusion coefficient at ~7 to 8 cm depths results from only one sample run. If one runs more samples and average them, I think the "unexpectedly" will be disappear. It is not from non-local mixing.'

Response:
We have addressed the issue of stochastic process in our response to Comment 2 by Anonymous Referee #2.

Discontinuous displacements in sediments and thus the biodiffusion coefficient are more expected from non-local mixing than local mixing (e.g., Shull, 2001). Thus, a sudden large displacement in relatively deep sediment can be reasonably attributed to non-local mixing. Nonetheless, after conducting multiple runs for each simulation, the sudden change described in the relevant sentence of the previous manuscript was found to occur in some runs but not in others.

Changes in manuscript (Page numbers/Line numbers):
We have added an appendix to show the contribution of stochastic process to the bioturbation results in the revised manuscript (Appendix D, P18/L25-29). Where necessary, we modified descriptions in

Section 3 of the revised manuscript (P12/L1-2, P12/L13-14, P12/L27-28, P13/L28, P14/L12, P15/L23-24). We removed the relevant sentence as it applies to some runs but not others (P12/L1-2).

Other note 7:

'12/3: The authors mentioned that "advective water flow has only insignificant influences on bioturbation". In fact, people are more interested in the effect of bioturbation on the advective water flow and thereafter the bioirrigation, which might significantly change oxygen flux.'

Response:

Currently, eLABS cannot explicitly simulate the pumping action of infauna, which we plan to implement in future model developments, as described in Section 4. Nonetheless, we ran a simulation where the water flows imposed at the time of ingestion/egestion are increased arbitrary by factors between $10^2$ and $10^4$ to implicitly include a bio-irrigation effect. With increasing biologically induced water flows, more strong oxygen-mixing can be caused by advection and the bioturbation results are similar to those in a simulation with a high shear velocity. Hence it is possible to address the importance of bioirrigation during bioturbation in such simulations with increased advective water flows.

Changes in manuscript (Page numbers/Line numbers):

We added an appendix which shows simulations where water advection caused by infauna at the time of ingestion/egestion is increased by factors between $10^2$ and $10^4$ (Appendix E, P19/L2-14). We reference this appendix in the relevant sentence (P13/L4-5).

---

## Author Response (AR2)

Concerning the paper entitled 'A lattice-automaton bioturbation simulator with coupled physics, chemistry, and biology in marine sediments (eLABS v0.2)' by Yoshiki Kanzaki, Bernard P. Boudreau, Sandra Kirtland Turner and Andy Ridgwell (gmd-2019-62)

September 19, 2019

Dear Dr. Guy Munhoven,

We express our gratitude to you for considering and accepting our paper for publication in GMD and to Referee#2 for another round of review. According to your comments, we revised Code availability section to describe DOIs of the model codes. Please see just below for the list of all relevant changes in the revised manuscript and further below for the marked-up manuscript.

List of all relevant changes

[revised manuscript text omitted]

Figure C1

[Figure]

Figure C2

[Figure]

Figure C3

[Figure]

Electric potential (V)

Figure D1

[Figure]

Figure E1

[Figure]

Figure E2

[Figure]

(a) High shear velocity

(b) Pump ×10²

(c) Pump ×10³

(d) Pump ×10⁴

25 days   50 days   100 days
150 days   200 days   250 days

12 cm

12 cm

0.0   0.2   0.4   0.6   0.8   1.0

Normalized O$_2$ concentration

Figure E3

[Figure]

Figure E4

[Figure]

(a) High shear velocity

| 25 days | 50 days | 100 days |
| 150 days | 200 days | 250 days |

(b) Pump ×10²

| 25 days | 50 days | 100 days |
| 150 days | 200 days | 250 days |

(c) Pump ×10³

| 25 days | 50 days | 100 days |
| 150 days | 200 days | 250 days |

(d) Pump ×10⁴

| 25 days | 50 days | 100 days |
| 150 days | 200 days | 250 days |

12 cm

12 cm

■ Sediment  □ Water  ■ Organism (head)  ■ Organism (body)

Figure E5

[Figure]